# DEL-Ranking: Ranking-Correction Denoising Framework for Elucidating Molecular Affinities in DNA-Encoded Libraries

## Abstract

DNA-encoded library (DEL) screening has revolutionized protein-ligand binding detection, enabling rapid exploration of vast chemical spaces through read count analysis. However, two critical challenges limit its effectiveness: distribution noise in low copy number regimes and systematic shifts between read counts and true binding affinities. We present DEL-Ranking, a comprehensive framework that simultaneously addresses both challenges through innovative ranking-based denoising and activity-referenced correction. Our approach introduces a dual-perspective ranking strategy combining Pair-wise Soft Rank (PSR) and List-wise Global Rank (LGR) constraints to preserve both local and global count relationships. Additionally, we develop an Activity-Referenced Correction (ARC) module that bridges the gap between read counts and binding affinities through iterative refinement and biological consistency enforcement. Another key contribution of this work is the curation and release of three comprehensive DEL datasets that uniquely combine ligand 2D sequences, 3D conformational information, and experimentally validated activity labels. We validate our framework on five diverse DEL datasets and introduce three new comprehensive datasets featuring 2D sequences, 3D structures, and activity labels. DEL-Ranking achieves state-of-the-art performance across multiple correlation metrics and demonstrates strong generalization ability across different protein targets. Importantly, our approach successfully identifies key functional groups associated with binding affinity, providing actionable insights for drug discovery. This work advances both the accuracy and interpretability of DEL screening, while contributing valuable datasets for future research.

## 1 Introduction

DNA-encoded library (DEL) technology has emerged as a revolutionary approach for protein-ligand binding detection, offering unprecedented advantages over traditional high-throughput screening methods(Franzini et al., 2014; Neri & Lerner, 2018; Peterson & Liu, 2023; Ma et al., 2023). The DEL screening process involves multiple stages including cycling, binding, washing, elution, and amplification (as shown in Figure 1). This process generates large-scale read count data, serving as a proxy for potential binding affinity(Machutta et al., 2017; Foley et al., 2021). The read counts represent the frequency of each compound in the selected pool after undergoing target protein binding and subsequent processing steps (Favalli et al., 2018). These read counts typically include matrix counts and target counts, representing values corresponding to the scenarios without and with specific protein targets, respectively.

DEL screening enables rapid and cost-effective evaluation of vast chemical spaces, typically encompassing billions of compounds (Satz et al., 2022), against biological targets (Neri & Lerner, 2017). This approach has gained widespread adoption in drug discovery due to its ability to identify novel chemical scaffolds and accelerate lead compound identification (Brenner & Lerner, 1992; Goodnow Jr & Davie, 2017; Yuen & Franzini, 2017). Despite its advantages, DEL screening faces two critical challenges: (1) DEL screening read counts are subject to **Distribution Noise**, stemming from both experimental factors and intrinsic library characteristics. This noise predominantly affects the low copy number regime ($<10$ copies) where read counts follow a Poisson distribution, creating

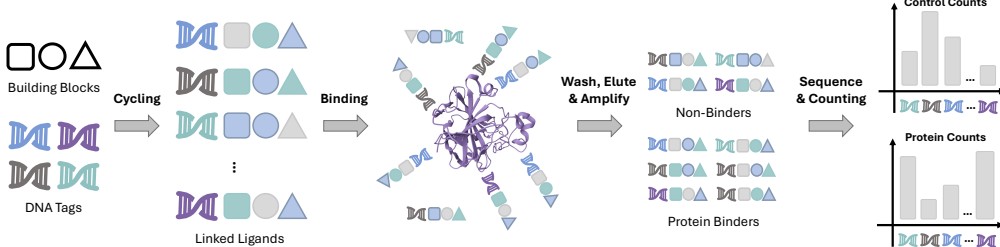

Figure 1: Illustration of the DEL screening process. **Cycling**: Creating unique compounds, each tagged with a distinctive DNA sequence. **Binding**: These compounds are then exposed to the target protein. **Wash, Elute and Amplify**: Compounds that bind to the target are retained, while others are washed away. The DNA tags of the bound compounds are then amplified and analyzed using sequencing techniques. **Sequence & Counting**: This process results in a distribution of read counts for both the target-bound samples and control samples.

significant discrepancies between observed counts and true binding affinities (Ki values) (Kuai et al., 2018; Favalli et al., 2018); and (2) DEL screening data faces **Distribution Shift**, where the mapping from read counts to binding affinities is systematically biased. While read counts reflect initial binding events, true binding affinity (Ki value) depends on multiple molecular properties that cannot be captured by enrichment data alone, creating a fundamental gap between read count-based and actual affinity distributions (Yung-Chi & Prusoff, 1973; Kuai et al., 2018). This study aims to address both challenges by enhancing the correlation between predicted read counts and true binding affinity.

To address these challenges, various approaches have been developed. Early methods focused on mitigating **Distribution Noise** through threshold-based filtering of enrichment factors calculated as target count to matrix count ratios (Gu et al., 2008; Kuai et al., 2018). While computationally efficient and interpretable, these methods only consider read count properties, ignoring the complex relationships between molecular structures and their corresponding counts (McCloskey et al., 2020). Recent advances have taken two main directions. First, machine learning approaches were introduced to capture non-linear relationships between ligand molecules and their count labels (McCloskey et al., 2020; Ma et al., 2021). This was further enhanced by incorporating distribution-level constraints, using ligand sequence embeddings to jointly predict enrichment factors and ensure count consistency (Lim et al., 2022; Hou et al., 2023). Second, recognizing the limitations of 2D representations, DEL-Dock (Shmilovich et al., 2023) introduced 3D conformational information to improve denoising. By incorporating Zero-Inflated Poisson distribution (ZIP) modeling with 3D structural information, DEL-Dock achieved superior performance in read count prediction and demonstrated strong generalization ability across different protein targets.

A critical yet often overlooked aspect in current approaches is the biological activity information inherent in molecular structures. While the presence of certain functional groups, such as **benzene sulfonamide**, can serve as reliable indicators of binding potential and provide complementary information to read counts (Hou et al., 2023; Blevins et al., 2024), and this prior knowledge can be extended to other DEL targets and datasets, current methods still face significant limitations in fully utilizing this information (Wichert et al., 2024). These methods primarily focus on absolute read count values instead of more robust relative ordering information, and rely solely on enrichment data without incorporating crucial biological activity labels. Although recent advanced computational approaches have made progress in addressing **Distribution Noise**, the fundamental challenge of **Distribution Shift** between enrichment data and true binding affinities remains largely unaddressed in current literature. This gap underscores the need for methods that can better integrate both functional group activity indicators and binding affinity information to improve both read count regression and the discovery of novel high-affinity functional groups.

To address these multifaceted challenges, we propose DEL-Ranking, an innovative framework that synergistically tackles both **Distribution Noise** and **Distribution Shift**. Our approach uniquely combines theoretically grounded ranking constraints with activity-referenced correction mecha-

nisms that leverage molecular structural features, enabling robust read count denoising while preserving biological relevance. At the core of DEL-Ranking's **Distribution Noise** mitigation is a novel dual-perspective ranking strategy. We introduce two complementary constraints: the **P**airwise **S**oft **R**ank (PSR) that captures local discriminative features, and the **L**ist-wise **G**lobal **R**ank (LGR) that maintains global distribution patterns. This ranking-based approach emphasizes relative relationships while preserving absolute count information, theoretically complementing the existing ZIP-based regression loss and reducing the expected error bound. To address **Distribution Shift**, we develop the **A**ctivity-**R**eferenced **C**orrection (ARC) module that bridges the gap between read counts and binding affinities. ARC operates through a two-stage process: a Refinement Stage that leverages self-training techniques (Zoph et al., 2020) for iterative optimization, and a Correction Stage that employs targeted consistency loss to ensure biological relevance. By incorporating novel activity labels derived from ligand functional group analysis, ARC effectively aligns read count predictions with true binding properties.

We validate our approach through extensive experiments on five diverse DEL datasets and introduce three new comprehensive datasets featuring ligand 2D sequences, 3D structures, and activity labels. Our method not only achieves state-of-the-art performance across multiple metrics but also successfully identifies key functional groups associated with high binding affinities. Through detailed ablation studies, we demonstrate each component's effectiveness and show how our activity-referenced framework can reveal structure-activity relationships that guide rational drug design. This work advances both DEL screening accuracy and interpretability, potentially accelerating drug discovery through more precise binding affinity predictions and actionable structural insights.

## 2 RELATED WORKS

Traditional DEL data analysis approaches, like QSAR models (Martin et al., 2017) and molecular docking simulations (Jiang et al., 2015; Wang et al., 2015), offer interpretability and mechanistic insights. DEL-specific methods such as data aggregation (Satz, 2016) and normalized z-score metrics (Faver et al., 2019) address unique DEL screening challenges. However, these methods face limitations in scalability and handling complex, non-linear relationships in large-scale DEL datasets.

Machine learning techniques such as Random Forest, Gradient Boosting Models, and Support Vector Machines have improved DEL data analysis (Li et al., 2018; Ballester & Mitchell, 2010). Combined with Bayesian Optimization (Hernández-Lobato et al., 2017), these methods offer better scalability and capture complex, non-linear relationships in high-dimensional DEL data. Despite outperforming traditional methods, they are limited by their reliance on extensive training data and lack of interpretability in complex biochemical systems.

Deep learning approaches, particularly Graph Neural Networks (GNNs), have significantly advanced protein-ligand interaction predictions in DEL screening. GNN-based models predict enrichment scores and accommodate technical variations (Stokes et al., 2020; Ma et al., 2021), while Graph Convolutional Neural Networks (GCNNs) enhance detection of complex molecular structures (McCloskey et al., 2020; Hou et al., 2023). Recent innovations include DEL-Dock, combining 3D pose information with 2D molecular fingerprints (Shmilovich et al., 2023), and sparse learning methods addressing noise from truncated products and sequencing errors (Kómár & Kalinic, 2020). Large-scale prospective studies have validated these AI-driven approaches, confirming improved hit rates and specific inhibitory activities against protein targets (Gu et al., 2024).

Existing methods demonstrate improved scalability and molecular interaction modeling. However, they face challenges in data interpretability and theoretical foundations. Current modeling approaches for DEL read count data primarily rely on theoretical prior distribution assumptions, lacking sophisticated noise handling mechanisms and robust statistical frameworks. These limitations manifest in the inability to incorporate information beyond prior distributions and insufficient reliable activity validation data for denoising, leading to suboptimal performance when addressing Distribution Noise and Distribution Shift in DEL data.

To address these limitations, we propose a novel denoising framework that integrates a theoretically grounded combined ranking loss with an iterative validation loop. This approach aims to correct read count distributions more effectively, addressing both Distribution Noise and Distribution Shift, while improving the interpretability and reliability of binding affinity predictions in DEL screening.

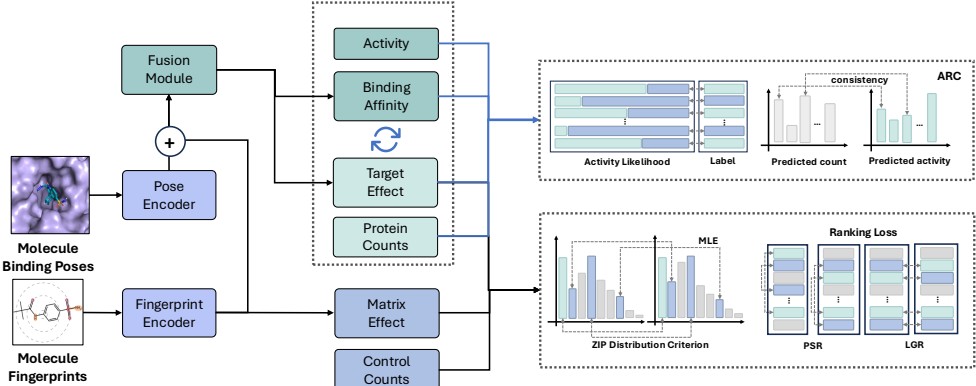

Figure 2: Overview of DEL-Ranking framework. The model directly fuses molecule binding poses and fingerprints as input features. ARC employs target effects and binding affinity to enhance read count prediction. The ranking-based loss incorporates target effects and matrix effects for noise removal, improving the correlation between predicted read counts and true binding affinities.

## 3 METHOD

We present DEL-Ranking framework that directly denoises DEL read count values and incorporates new activity information. In Section 3.1, we directly formulate the problem. Sections 3.2 and 3.3 detail our innovative modules, while Section 3.4 introduces the overall training objective.

### 3.1 PROBLEM FORMULATION AND PRELIMINARIES

**DEL Prediction Framework.** Given a DEL dataset $\mathcal{D} = \{(\mathbf{f}_i, \mathbf{p}_i, M_i, R_i, y_i)\}_{i=1}^N$, where $\mathbf{f}_i \in \mathbb{R}^d$ denotes the molecular fingerprint, $\mathbf{p}_i \in \mathbb{R}^m$ represents the binding pose, $M_i \in \mathbb{R}$ is the matrix count derived from control experiments without protein targets, $R_i \in \mathbb{R}$ is the target count obtained from experiments involving protein target binding, and $y_i \in \{0, 1\}$ indicates the activity label. We propose a joint multi-task learning framework $\mathcal{F} : \mathbb{R}^d \times \mathbb{R}^m \to \mathbb{R} \times \mathbb{R} \times [0, 1]$ such that:

$$\mathcal{F}(\mathbf{f}_i, \mathbf{p}_i) = (\hat{M}_i, \hat{R}_i, \hat{p}_i) \tag{1}$$

where $\hat{M}_i$ represents the predicted matrix count, $\hat{R}_i$ denotes the predicted target count, and $\hat{p}_i$ is the predicted activity likelihood. The primary focus of this framework lies in predicting accurate read count values that strongly correlate with the actual $K_i$ values.

**Zero-Inflated Poisson Distribution (ZIP) & ZIP Loss.** DEL screening often results in read count distributions with a high proportion of zeros due to experimental factors. To address this, previous methods (Shmilovich et al., 2023; Lim et al., 2022) have employed zero-inflated distributions, modeling read counts $r_i$ that can take values from either from target counts $M_i$ or target counts $R_i$. Similarly, we define $\hat{r}_i \in \{\hat{M}_i, \hat{R}_i\}$ as the model's predicted read count values.

$$P(X = r_i | \lambda, \pi) = \begin{cases} \pi + (1 - \pi)e^{-\lambda}, & \text{if } r_i = 0 \\ (1 - \pi)\frac{\lambda_i^r e^{-\lambda}}{r_i!}, & \text{if } r_i > 0 \end{cases} \tag{2}$$

where $\pi$ denotes the probability of excess zeros, and $\lambda$ denotes the mean parameter of the Poisson component. In (Shmilovich et al., 2023), ZIP of $M_i$, $R_i$ are modeled respectively by different $\pi$ values, including $\pi_M$ and $\pi_R$, based on orders of magnitude, with the regression achieved by minimizing the Negative Log-Likelihood (NLL) for all predicted read counts $\hat{M}_i$ and $\hat{R}_j$.

$$\mathcal{L}_{\text{ZIP}} = -\sum_i \log[P(\hat{M}_i | \lambda_M, \pi_M)] - \sum_j \log[P(\hat{R}_j | \lambda_M + \lambda_R, \pi_R)] \tag{3}$$

where $\lambda_M$ and $\lambda_R$ represent Poisson mean parameters for matrix and target counts, and $\pi_M$ and $\pi_R$ denote their zero-excess probabilities. Joint modeling of target and control counts captures the

relationship between DEL experiments while enhancing the model's understanding of read count correlations with inputs.

$K_i$ **Estimation.** DEL read count prediction aims to estimate compound-target binding affinities – experimental-valided $K_i$ values, which are crucial for identifying promising drug candidates. We evaluate our model's effectiveness using the Spearman rank correlation coefficient ($\rho_s$) between predicted read counts and true $K_i$ values: $\rho_s = 1 - \frac{6 \sum d^2}{n(n^2-1)}$ where $n$ is the sample number and $d$ is the predicted ranking discrepancy. Ideally, $K_i$ values and read counts are negatively correlated.

## 3.2 RANKING-BASED DISTRIBUTION NOISE REMOVAL

To effectively remove **Distribution Noise** in DEL read count data, we propose a novel ranking-based loss function $\mathcal{L}_{\text{rank}}$. This loss function integrates both local and global read count information to achieve a well-ordered Zero-Inflated Poisson distribution for read count values:

$$\mathcal{L}_{\text{rank}} = \beta \mathcal{L}_{\text{PSR}} + (1 - \beta)\mathcal{L}_{\text{LGR}} \tag{4}$$

where $\beta \in [0, 1]$ is a balancing hyperparameter to fit the relative magnitude of two components. $\mathcal{L}_{\text{PSR}}$ (Pairwise-Soft Ranking Loss) addresses local pairwise comparisons, while $\mathcal{L}_{\text{LGR}}$ (listwise Global Ranking Loss) captures global ranking information. Together, they aim to achieve a well-ordered Zero-Inflated Poisson distribution for read count values. To formally establish the effectiveness of our ranking-based approach, we provide the following theoretical justification:

**Lemma 1.** *Given a set of feature-read count pairs $\{(x_i, r_i)\}_{i=1}^n$, where $x_i$ is the fused representation of sample $i$ based on $f_i$ and $p_i$, and a well-fitted Zero-Inflated Poisson model $f_{\text{ZIP}}(r|x)$, the ranking loss $\mathcal{L}_{\text{rank}}$ provides positive information gain over the zero-inflated loss $\mathcal{L}_{\text{ZIP}}$:*

$$I(\mathcal{L}_{rank}|\mathcal{L}_{\text{ZIP}}) = H(R|\mathcal{L}_{\text{ZIP}}) - H(R|\mathcal{L}_{\text{ZIP}}, \mathcal{L}_{rank}) > 0$$

*where $H(R|\cdot)$ denotes the conditional entropy of read counts $R$.*

Building upon this information gain, we can further demonstrate that our combined approach, which incorporates both the zero-inflated and ranking losses, outperforms the standard zero-inflated model in terms of expected loss. This improvement is formalized in the following theorem:

**Theorem 2.** *Given a sufficiently large dataset $\{(x_i, r_i)\}_{i=1}^n$ of feature-read count pairs, let $\mathcal{L}_{\text{ZIP}}$ be the loss function of standard zero-inflated model and $\mathcal{L}_{rank}$ be the combined ranking loss. For predictions $\hat{r}^{ZI}$ and $\hat{r}^C$ from the standard and combined models respectively. Define $\mathcal{L}_C = \alpha\mathcal{L}_{\text{ZIP}} + (1 - \alpha)\mathcal{L}_{rank}$, there exists $\alpha \in [0, 1]$ such that:*

$$E[\mathcal{L}_C(\hat{r}^C)] < E[\mathcal{L}_{\text{ZIP}}(\hat{r}^{ZI})]$$

The incorporated ranking information aligns read count across compounds, mitigating experimental biases in DEL screening data. Detailed proof and analysis are provided in Appendices A.1 and A.2.

### 3.2.1 PAIRWISE SOFT RANKING LOSS

To better model the relationships between compound pairs and handle read count noise, we introduce a novel Pairwise Soft Ranking (PSR) loss function. The PSR loss enables smooth comparison between compounds while maintaining stable optimization. $\mathcal{L}_{\text{PSR}}$ is defined as:

$$\mathcal{L}_{\text{PSR}}(\hat{r}_i, \hat{r}_j, T) = -\sum_{i=1}^n \hat{r}_i \left( \sum_{j\neq i}(\Delta_{ij} \cdot \sigma_{ij}) - \sum_{j\neq i}(\Delta_{ji} \cdot \sigma_{ji}) \right)$$

$$\sigma_{ij} = \frac{1}{1 + e^{-|r_i-r_j|/T}}, \quad \Delta_{ij} = \frac{\Delta G_{ij} \cdot \Delta D_{ij}}{Z} \tag{5}$$

where $\hat{r}_i$ and $\hat{r}_j$ represent the predicted read count value for compound $i$ and $j$, respectively. To ensure smooth gradients and numerical stability, we introduce a scaling factor $\sigma_{ij}$ with temperature $T$ ensures smooth transitions between rankings. To accurately model ranking changes, we design a pairwise importance term $\Delta_{ij}$ that quantifies the impact of swapping compounds $i$ and $j$. This term incorporates a gain function $G_i = \text{softplus}(r_i)$ that reflects the relevance of each compound,

and a rank-based discount function $D_i = 1/(\log_2(\text{rank}_i + 1) + \epsilon)$, where $\epsilon$ is a small constant for numerical stability. These gain and discount mechanisms emphasize high-ranking samples, enabling the model to focus on top-ranking cases and effectively identify potentially high-activity compounds. By combining these components into smooth Delta functions, we ensure stable gradient flow:

$$\Delta G_{ij} = G_i - G_j = \text{softplus}(r_i) - \text{softplus}(r_j)$$
$$\Delta D_{ij} = D_i - D_j = \frac{1}{(\log_2(\text{rank}_i + 1) + \epsilon)} - \frac{1}{(\log_2(\text{rank}_j + 1) + \epsilon)} \tag{6}$$

To normalize ranking effects, we introduce a normalization factor computed from the top-K predicted values in each batch. By choosing K smaller than the batch size N, we achieve two benefits: enhanced computational efficiency and avoidance of ranking noise from zero-value predictions.

$$Z = \sum_{k=1}^{K} \frac{\text{softplus}(\hat{r}_{[k]})}{\log_2(k + 1) + \epsilon} \tag{7}$$

where $\hat{r}_{[k]}$ represents the k-th highest predicted read count in descending order; $\epsilon$ is set to $1e^{-8}$ to avoid division by zero. This normalization factor adjusts the loss scale across different dataset sizes and read count distributions, ensuring robust model training regardless of data variations.

### 3.2.2 LISTWISE GLOBAL RANKING LOSS

To further consider global order in compound ranking, we propose the listwise Global Ranking (LGR) loss $\mathcal{L}_{\text{LGR}}$ as a complement to the Pairwise Soft Ranking loss $\mathcal{L}_{\text{PSR}}$, which is expressed as :

$$\mathcal{L}_{\text{LGR}}(\hat{r}, \tau, T) = -\sum_{i=1}^{N} \log \frac{\exp(\hat{r}_{\pi(i)}/T)}{\sum_{j=i}^{N} \exp(\hat{r}_{\pi(j)}/T)} + \sigma \sum_{i=1}^{N} \sum_{j \neq i} \mathcal{L}_{\text{con}}(\hat{r}_i, \hat{r}_j, \tau) \tag{8}$$

where $\pi$ is the true ranking permutation of the compounds; $\tau$ represents the minimal margin of predicted read count pair $(r_i, r_j)$; $T$ is a temperature parameter for score rescaling to sharpen the predicted distribution, and $\mathcal{L}_{\text{con}}$ denotes a contrastive loss among ranking scores to capture local connections, and $\sigma$ denotes the weight. This formulation is designed to achieve two critical objectives in DEL experiments: (1) **Near-deterministic selection** of compounds with the highest read counts, corresponding to the highest binding affinities; (2) **Increased robustness** to small noise perturbations in the experimental data. As $T$ approaches 0, our model becomes increasingly selective towards high-affinity compounds while maintaining resilience against common experimental noises. This dual optimization leads to more consistent identification of promising drug candidates and enhanced reliability in the face of experimental variability.

Despite the strengths of $\mathcal{L}_{\text{PSR}}$ and $\mathcal{L}_{\text{LGR}}$, they struggle to differentiate activity levels among compounds with identical read count values, particularly those affected by experimental noise. This limitation can lead to misclassification of high-activity samples with artificially low read counts as truly low-activity samples. To address this critical issue, we introduce a novel contrastive loss function $\mathcal{L}_{\text{con}}$, designed to enhance discrimination between varying levels of biological activity, especially for samples with zero or identical read count values. Let $f : \mathcal{R} \to \mathbb{R}$ be a ranking function and $\tau > 0$ a fixed threshold. We define $\mathcal{L}_{\text{con}} : \mathcal{R} \times \mathcal{R} \times \mathcal{R} \to \mathbb{R} \geq 0$ as:

$$\mathcal{L}_{\text{con}}(\hat{r}_i, \hat{r}_j, \tau) = \max\{0, \tau - (f(\hat{r}_i) - f(\hat{r}_j))\} \tag{9}$$

This loss function is positive if and only if $f(\hat{r}_i) - f(\hat{r}_j) < \tau$, enforcing a minimum margin $\tau$ between differently ranked samples. The constant gradients $\partial \mathcal{L}_{\text{con}}/\partial f(\hat{r}_i) = -1$ and $\partial \mathcal{L}_{\text{con}}/\partial f(\hat{r}_j) = 1$ for $f(\hat{r}_i) - f(\hat{r}_j) < \tau$ promote robust ranking relationships.

### 3.3 ACTIVITY-REFERENCED DISTRIBUTION CORRECTION FRAMEWORK

To effectively leverage activity information and address **Distribution Shifts** in DEL experiments, we propose the **A**ctivity-**R**eferenced **C**orrection (ARC) framework. This algorithm enhances read count distribution fitting through two complementary stages: the Refinement Stage, which integrates activity information, and the Consistency Stage, which performs distribution adjustment, jointly optimizing the fitting from both dimensions. The Refinement Stage leverages dual information streams

**Algorithm 1** Refinement Stage for Activity-Referenced Correction (ARC) Algorithm

---

**Require:** Pose structure embeddings $\mathbf{h}_p$, Fingerprint sequence embeddings $\mathbf{h}_f$, Sequence-structure balancing weight $\varsigma$, num_iterations $\mathbf{n}$, use_feedback

1: $x \leftarrow \texttt{PostAddLayer}(\varsigma \mathbf{h}_p + \mathbf{h}_f)$
2: $\hat{M} \leftarrow \texttt{MatrixHead}(\mathbf{h}_f)$
3: Initialize $\hat{R} \leftarrow \mathbf{0}, \hat{y} \leftarrow \mathbf{0}$
4: **for** $i = 1$ to $\mathbf{n}$ **do**
5:     **if** use_feedback **then**
6:         $\hat{R} \leftarrow [x; \hat{p}], \hat{p} \leftarrow [x; \hat{R}]$
7:     **else**
8:         $\hat{R} \leftarrow x, \hat{p} \leftarrow x$
9:     **end if**
10:     $\hat{R} \leftarrow \texttt{EnrichmentHead}(\texttt{ReadHead}(\hat{R}))$
11:     $\hat{p} \leftarrow \texttt{ActHead}(\hat{p})$
12: **end for**
      **return** $\hat{M}, \hat{R}, \hat{p}$

---

- activity labels and read counts - which reflect compound activity from complementary angles: activity labels capture overall binding potential, while read counts quantify binding strength. As detailed in Algorithm 1, our approach generates initial predictions using both 2D SMILES embeddings and 2D-3D joint embeddings for matrix and target counts. To fuse these predictions effectively, we implement an adaptive iterative mechanism inspired by self-training techniques (Zoph et al., 2020). Through multiple rounds of updates, a bidirectional feedback loop is established: activity information calibrates noisy read count predictions, while read count patterns help validate activity predictions. This iterative refinement ensures biological consistency while enhancing prediction accuracy through mutual correction between both information spaces.

To address error accumulation and better align predictions with biological reality, we introduce a consistency loss function in the Correction Stage. This function not only regresses predicted values but also aligns prediction trends with activity labels. Following (Hou et al., 2023), we define ground-truth labels based on the presence of benzene sulfonamide in molecules. Although there may exist high-affinity molecules without benzene sulfonamide, this labeling scheme provides effective supervision signals for both read count regression and novel high-affinity functional group discovery, as demonstrated in Section 4.2. This approach helps resolve discrepancies in compounds that exhibit low read counts but high activity. The consistency loss is defined as:

$$\mathcal{L}_{\text{consist}}(r_i, \hat{r}_i, y_i, \hat{y}_i) = \|\hat{y}_i - y_i\| + \max\left(0, \|\hat{y}_i - \frac{\hat{r}_i}{\max_{i \in \{1, \ldots, N\}} \hat{r}_i}\|_2^2 - \|y_i - \frac{r_i}{\max_{i \in \{1, \ldots, N\}} r_i}\|_2^2\right) \tag{10}$$

where N denotes training batch size. The first term ensures prediction accuracy, while the second term constrains the consistency between read count values and binary activity predictions.

### 3.4 TOTAL TRAINING OBJECTIVE

The total training objective integrates three distinct components. The Zero-Inflated Poisson distribution loss $\mathcal{L}_{\text{ZIP}}$ models the overall read count distribution, while the combined ranking loss $\mathcal{L}_{\text{rank}}$ refines the predicted ZIP distribution based on ordinal relationships. Additionally, the consistency loss $\mathcal{L}_{\text{consist}}$ further adjusts the distribution using activity labels. These components are combined into the total loss function as follows:

$$\mathcal{L}_{\text{total}} = \mathcal{L}_{\text{ZIP}} + \rho \mathcal{L}_{\text{rank}} + \gamma \mathcal{L}_{\text{consist}} \tag{11}$$

where $\rho$ and $\gamma$ are weighting factors for the ranking and consistency losses, respectively.

## 4 EXPERIMENT

**Datasets. CA9 Dataset** From the original data containing 108,529 DNA-barcoded molecules targeting human carbonic anhydrase IX (CA9) (Gerry et al., 2019), we derived two separate datasets.

Table 1: Comparison of our framework DEL-Ranking with existing DEL affinity predictions on CA2 & CA12 datasets. Results in **bold** and underlined are the top-1 and top-2 performances, respectively.

| Metric | 3p3h (CA2) Sp | SubSp | 4kp5-A (CA12) Sp | SubSp | 4kp5-OA (CA12) Sp | SubSp |
|---|---|---|---|---|---|---|
| Mol Weight | -0.250 | -0.125 | -0.101 | 0.020 | -0.101 | 0.020 |
| Benzene | 0.022 | 0.072 | -0.054 | 0.035 | -0.054 | 0.035 |
| Vina Docking | $-0.174_{\pm0.002}$ | $-0.017_{\pm0.003}$ | $0.025_{\pm0.001}$ | $0.150_{\pm0.003}$ | $0.025_{\pm0.001}$ | $0.150_{\pm0.003}$ |
| RF-Enrichment | $-0.017_{\pm0.026}$ | $-0.042_{\pm0.025}$ | $-0.029_{\pm0.038}$ | $-0.005_{\pm0.048}$ | $\underline{-0.101_{\pm0.009}}$ | $-0.087_{\pm0.010}$ |
| RF-ZIP | $0.027_{\pm0.139}$ | $-0.005_{\pm0.071}$ | $0.035_{\pm0.094}$ | $-0.026_{\pm0.111}$ | $0.006_{\pm0.095}$ | $-0.021_{\pm0.122}$ |
| Dos-DEL | $-0.048_{\pm0.036}$ | $-0.011_{\pm0.035}$ | $-0.016_{\pm0.029}$ | $-0.017_{\pm0.021}$ | $-0.003_{\pm0.030}$ | $-0.048_{\pm0.034}$ |
| DEL-QSVR | $-0.228_{\pm0.021}$ | $\underline{-0.171_{\pm0.033}}$ | $-0.004_{\pm0.178}$ | $0.018_{\pm0.139}$ | $0.070_{\pm0.134}$ | $-0.076_{\pm0.116}$ |
| DEL-Dock | $\underline{-0.255_{\pm0.009}}$ | $-0.137_{\pm0.012}$ | $\underline{-0.242_{\pm0.011}}$ | $\underline{-0.263_{\pm0.012}}$ | $0.015_{\pm0.029}$ | $\underline{-0.105_{\pm0.034}}$ |
| DEL-Ranking | $\mathbf{-0.286_{\pm0.002}}$ | $\mathbf{-0.177_{\pm0.005}}$ | $\mathbf{-0.268_{\pm0.012}}$ | $\mathbf{-0.277_{\pm0.016}}$ | $\mathbf{-0.289_{\pm0.025}}$ | $\mathbf{-0.233_{\pm0.021}}$ |

The first, denoted as **5fl4-9p**, uses 9 docked poses that we generated ourselves. The second, **5fl4-20p**, employs 20 docked poses using the 5fl4 structure. Both datasets lack activity labels. **CA2 and CA12 Datasets** From the CAS-DEL library (Hou et al., 2023), we generated three datasets comprising 78,390 molecules selected from 7,721,415 3-cycle peptide compounds. We performed docking to create 9 poses per molecule for each dataset. The CA2-derived dataset uses the 3p3h PDB structure (denoted as **3p3h**), while two CA12-derived datasets use the 4kp5 PDB structure: **4kp5-A** for normal expression and **4kp5-OA** for overexpression. The binary activity label is set to 1 when there is benzene sulfonamide (BB3-197) in the compound (Hou et al., 2023). **Validation Dataset** from ChEMBL (Zdrazil et al., 2024) includes 12,409 small molecules with affinity measurements for CA9, CA2, and CA12. Molecules have compatible atom types, molecular weights from 25 to 1000 amu, and inhibitory constants ($K_i$) from 90 pM to 0.15 M. A subset focusing on the 10-90th percentile range of the training data's molecular weights provides a more challenging test scenario. **Virtual Docking** details for ligand poses are shown in Appendix B.1.

**Evaluation Metrics and hyper-parameters.** To evaluate our framework's effectiveness, we employ two Spearman correlation metrics on the ChEMBL dataset (Zdrazil et al., 2024). The first metric, overall Spearman correlation ($\rho_{\mathrm{overall}}$), measures the correlation between predicted read counts and experimentally determined $K_i$ values across the entire validation dataset. Also, we utilize the subset Spearman correlation ($\rho_{\mathrm{subset}}$), which focuses on compounds with molecular weights within the 10th to 90th percentile range of the training dataset. Since multiple hyper-parameters are shown in DEL-Ranking method, we provide a detailed hyper-parameter ssetting in Appendix B.2.

**Baselines.** We examine the performance of existing binding affinity predictors. Traditional methods based on binding poses and fingerprints include Molecule Weight, Benzene Sulfonamide, Vina Docking (Koes et al., 2013), and Dos-DEL(Gerry et al., 2019). AI-aided methods dependent on read count values and molecule information include RF-Enrichment, RF-ZIP (Random Forest for Log-enrichment, $\mathcal{L}_{\mathrm{ZIP}}$), DEL-QSVR, and DEL-Dock (Lim et al., 2022; Shmilovich et al., 2023).

## 4.1 Performance Comparison

**Benchmark Comparison.** We conducted comprehensive experiments across five diverse datasets: 3p3h, 4kp5-A, 4kp5-OA, and two variants of 5fl4. For each dataset, we performed five runs to ensure statistical robustness. As shown in Table 1-2, our method consistently achieves state-of-the-art results in both Spearman (**Sp**) and subset Spearman (**SubSp**) coefficients across all datasets.

Our analysis reveals several key insights: (1) **Experimental Adaptability**: DEL-Ranking shows consistent advantages across diverse datasets, with notable gains in challenging conditions. It maintains improvements even in lower-noise environments like purified protein datasets (3p3h and 5fl4), versatility highlighting DEL-Ranking's adaptability to various experimental setups. (2) **Noise Resilience**: DEL-Ranking excels in high-noise scenarios, particularly in membrane protein experiments. Its exceptional results on the 4kp5 dataset, especially the challenging 4kp5-OA variant, demonstrate this. Where baseline methods struggle, our approach effectively distinguishes signal from noise in complex experimental conditions. (3) **Structural Flexibility**: Our approach effectively uses structural information, as shown in the 5fl4 dataset. Increasing poses from 9 to 20 im-

Table 2: Comparison of our framework DEL-Ranking with existing DEL affinity predictions on two CA9 datasets. Results in **bold** and underlined are the top-1 and top-2 performances, respectively.

| | 5fl4-9p (CA9) | | 5fl4-20p (CA9) | |
|---|---|---|---|---|
| Metric | Sp | SubSp | Sp | SubSp |
| Mol Weight | -0.121 | -0.028 | -0.121 | -0.074 |
| Benzene | -0.174 | -0.134 | -0.199 | -0.063 |
| Vina Docking | $-0.114_{\pm 0.009}$ | $-0.055_{\pm 0.007}$ | $-0.279_{\pm 0.044}$ | $-0.091_{\pm 0.061}$ |
| RF-Enrichment | $-0.064_{\pm 0.126}$ | $-0.144_{\pm 0.024}$ | $-0.064_{\pm 0.126}$ | $-0.144_{\pm 0.024}$ |
| RF-ZIP | $0.040_{\pm 0.022}$ | $-0.011_{\pm 0.042}$ | $0.054_{\pm 0.094}$ | $0.026_{\pm 0.111}$ |
| Dos-DEL | $-0.115_{\pm 0.065}$ | $-0.036_{\pm 0.010}$ | $-0.231_{\pm 0.007}$ | $-0.091_{\pm 0.012}$ |
| DEL-QSVR | $-0.086_{\pm 0.060}$ | $-0.036_{\pm 0.074}$ | $-0.298_{\pm 0.005}$ | $-0.075_{\pm 0.011}$ |
| DEL-Dock | $\underline{-0.308_{\pm 0.000}}$ | $\underline{-0.169_{\pm 0.000}}$ | $\underline{-0.320_{\pm 0.009}}$ | $\underline{-0.166_{\pm 0.017}}$ |
| **DEL-Ranking** | $\mathbf{-0.323_{\pm 0.015}}$ | $\mathbf{-0.175_{\pm 0.000}}$ | $\mathbf{-0.330_{\pm 0.007}}$ | $\mathbf{-0.187_{\pm 0.013}}$ |

proves model performance, highlighting our method's ability to utilize additional structural data. This underscores DEL-Ranking's effectiveness in extracting insights from comprehensive structural information. (4) **Dual Analysis Capability**: DEL-Ranking's consistent performance in both Sp and SubSp metrics shows its versatility in drug discovery. This enables effective broad-spectrum screening and detailed subset analysis, enhancing its utility across various stages of drug discovery.

**Zero-shot Generalization.** We evaluated models' zero-shot generalization on CA9 by training them on CA2 and CA12 targets across three datasets (3p3h, 4kp5-A, and 4kp5-OA). Detailed in Table 3, DEL-Ranking consistently outperformed DEL-Dock. Notably, on the 4kp5-OA dataset with substantially different protein targets, DEL-Ranking maintained strong predictive performance, demonstrating its generalization capability to novel targets. Interestingly, DEL-QSVR exhibited superior zero-shot performance, suggesting that simpler molecular representations and loss functions might be more conducive to target generalization. This superior performance might be attributed to the fact that incorporating pose information could potentially limit zero-shot generalization capability.

## 4.2 DISCOVERY OF POTENTIAL HIGH AFFINITY FUNCTIONAL GROUP

To evaluate DEL-Ranking's capability in identifying potent compounds, we analyzed the Top-50 cases from our model across five datasets. Detailed in Figure 3, selected compounds consistently exhibited low $K_i$ values, revealing the model's ability to prioritize high-affinity compounds from large DEL libraries.

**Known Group Accuracy.** DEL-Ranking shows expectational accuracy in detecting benzene sulfonamide, a key high-affinity group for carbonic anhydrase inhibitors (Hou et al., 2023). From Figure 3, the model achieved high detection rates on five datasets, demonstrating that our ARC framework effectively incorporates biological prior knowledge into model prediction. To further explore the potential high-affinity compounds, we conducted the same study of DEL-Dock (Shmilovich et al., 2023) in Appendix C.3.

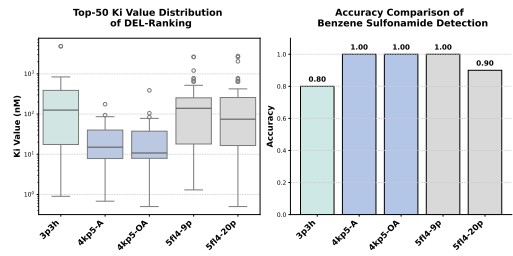

Figure 3: Quantitative analysis of Top-50 selection, including $K_i$ distribution and accuracy.

**Novel Group Discovery** Our analysis of the 3p3h and 5fl4 datasets revealed a significant finding: 20% (10/50) of high-ranking compounds in 3p3h and 10% (5/50) in 5fl4 lack the expected benzene sulfonamide group. Remarkably, all these compounds contain a common functional group - **Pyrimidine Sulfonamide** - which shares high structural similarity with benzene sulfonamide.

Further investigation through case-by-case Ki value determination yielded compelling results. Five compounds from 3p3h and five from 5fl4 containing pyrimidine sulfonamide exhibited $K_i$ values comparable to or even surpassing those of benzene sulfonamide-containing compounds. This finding profoundly validates DEL-Ranking's dual capability: successfully incorporating activity label information, while simultaneously leveraging multi-level information along with integrated ranking

Table 3: Zero-shot Generalization Results Comparison evaluated on CA9 dataset

| | 3p3h (CA2) | | 4kp5-A (CA12) | | 4kp5-OA (CA12) | |
|---|---|---|---|---|---|---|
| Metric | Sp | SubSp | Sp | SubSp | Sp | SubSp |
| Mol Weight | -0.121 | -0.028 | -0.121 | -0.028 | -0.121 | -0.028 |
| Benzene | -0.174 | -0.134 | -0.174 | -0.134 | -0.174 | -0.134 |
| Vina Docking | $-0.114_{\pm 0.009}$ | $-0.055_{\pm 0.007}$ | $-0.114_{\pm 0.009}$ | $-0.055_{\pm 0.007}$ | $-0.114_{\pm 0.009}$ | $-0.055_{\pm 0.007}$ |
| RF-Enrichment | $0.020_{\pm 0.014}$ | $-0.031_{\pm 0.057}$ | $-0.034_{\pm 0.013}$ | $-0.034_{\pm 0.029}$ | $-0.044_{\pm 0.005}$ | $-0.085_{\pm 0.006}$ |
| RF-ZIP | $0.037_{\pm 0.059}$ | $0.013_{\pm 0.017}$ | $0.036_{\pm 0.024}$ | $-0.002_{\pm 0.016}$ | $0.049_{\pm 0.012}$ | $-0.007_{\pm 0.013}$ |
| Dos-DEL | $-0.115_{\pm 0.065}$ | $-0.036_{\pm 0.010}$ | $-0.115_{\pm 0.065}$ | $-0.036_{\pm 0.010}$ | $-0.115_{\pm 0.065}$ | $-0.036_{\pm 0.010}$ |
| DEL-QSVR | $-0.300_{\pm 0.020}$ | $\mathbf{-0.257_{\pm 0.022}}$ | $\mathbf{-0.236_{\pm 0.038}}$ | $\mathbf{-0.223_{\pm 0.030}}$ | $0.108_{\pm 0.089}$ | $0.130_{\pm 0.070}$ |
| DEL-Dock | $-0.272_{\pm 0.013}$ | $-0.118_{\pm 0.005}$ | $-0.211_{\pm 0.007}$ | $-0.118_{\pm 0.010}$ | $0.065_{\pm 0.021}$ | $-0.125_{\pm 0.034}$ |
| **DEL-Ranking** | $\mathbf{-0.310_{\pm 0.005}}$ | $-0.120_{\pm 0.011}$ | $-0.228_{\pm 0.010}$ | $-0.127_{\pm 0.018}$ | $\mathbf{-0.300_{\pm 0.026}}$ | $\mathbf{-0.129_{\pm 0.021}}$ |

orders to uncover potential high-activity functional groups. Notably, this discovery reveals DEL-Ranking's ability to identify unexplored scaffolds, showing potential to improve compound prioritization and accelerate hit-to-lead optimization in early-stage drug discovery. Detailed visualization of Top-50 samples and selected Pyrimidine Sulfonamide cases are shown in Appendices C.3 and C.5.

## 4.3 ABLATION STUDY

To further explore the effectiveness of our enhancement, we compare DEL-Ranking with some variants on 3p3h, 4kp5-A, and 4kp5-OA datasets. We can observe from Table 4 that (1) $\mathcal{L}_{\text{PSR}}$ and $\mathcal{L}_{\text{LGR}}$ contribute most significantly to model performance across all datasets. (2) The impact of $\mathcal{L}_{\text{PSR}}$ is more pronounced in datasets with higher noise levels, as evidenced by the larger relative performance drop in the 3p3h dataset. (3) Temperature adjustment and $\mathcal{L}_{\text{consist}}$ help improve the performance by correcting the predicted distributions, but count less than ranking-based denoising.

Table 4: Ablation Study Results of DEL-Ranking on 3p3h, 4kp5-A, and 4kp5-OA datasets.

| | 3p3h (CA2) | | 4kp5-A (CA12) | | 4kp5-OA (CA12) | |
|---|---|---|---|---|---|---|
| Metric | Sp | SubSp | Sp | SubSp | Sp | SubSp |
| w/o All | $-0.255_{\pm 0.004}$ | $-0.137_{\pm 0.012}$ | $-0.242_{\pm 0.011}$ | $-0.263_{\pm 0.012}$ | $0.015_{\pm 0.029}$ | $-0.105_{\pm 0.034}$ |
| w/o $\mathcal{L}_{\text{PSR}}$ | $-0.273_{\pm 0.012}$ | $-0.155_{\pm 0.013}$ | $-0.251_{\pm 0.015}$ | $-0.271_{\pm 0.011}$ | $0.015_{\pm 0.028}$ | $-0.105_{\pm 0.033}$ |
| w/o $\mathcal{L}_{\text{LGR}}$ | $-0.280_{\pm 0.011}$ | $-0.168_{\pm 0.015}$ | $-0.256_{\pm 0.023}$ | $-0.273_{\pm 0.016}$ | $-0.269_{\pm 0.024}$ | $-0.209_{\pm 0.034}$ |
| w/o $\mathcal{L}_{\text{con}}$ | $-0.283_{\pm 0.004}$ | $-0.172_{\pm 0.007}$ | $-0.260_{\pm 0.018}$ | $-0.273_{\pm 0.014}$ | $-0.273_{\pm 0.024}$ | $-0.218_{\pm 0.034}$ |
| w/o Temp | $-0.279_{\pm 0.011}$ | $-0.166_{\pm 0.015}$ | $-0.247_{\pm 0.022}$ | $-0.265_{\pm 0.014}$ | $-0.256_{\pm 0.033}$ | $-0.181_{\pm 0.046}$ |
| w/o ARC | $-0.284_{\pm 0.007}$ | $-0.174_{\pm 0.010}$ | $-0.260_{\pm 0.015}$ | $-0.272_{\pm 0.012}$ | $-0.269_{\pm 0.023}$ | $-0.223_{\pm 0.045}$ |
| **DEL-Ranking** | $-0.286_{\pm 0.002}$ | $-0.177_{\pm 0.005}$ | $-0.268_{\pm 0.012}$ | $-0.277_{\pm 0.016}$ | $-0.289_{\pm 0.025}$ | $-0.233_{\pm 0.021}$ |

To further validate the robustness of DEL-Ranking and the incorporation of additional pose 3D structure information, we conducted ablation studies on both loss weight and structure information weight. The detailed results can be found in Appendices C.1 and C.2. The experimental results corroborate the capability of our approach and the feasibility of our hyperparameter selection criteria.

## 5 CONCLUSION

In this paper, we propose DEL-Ranking to addresses the challenge of noise in DEL screening through innovative ranking loss and activity-based correction algorithms. Experimental results demonstrate significant improvements in binding affinity prediction and generalization capability. Besides, the ability to identify potential binding affinity determinants advances the field of DEL screening analysis. Current limitations revolve around the challenges of acquiring, integrating, and comprehensively analyzing high-quality multi-modal molecular data at scale. Future works will aim to improve multi-modal data integration and analysis to advance DEL-based drug discovery.

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

# A THEORETICAL ANALYSIS

## A.1 PROOF OF LEMMA AND THEOREM

*Proof.* **[Proof of Lemma1]** Let $(\Omega, \mathcal{F}, P)$ be a probability space and $(X, Y) : \Omega \to \mathcal{X} \times \mathbb{N}_0$ be random variables representing features and read counts respectively. Define $f_{\text{ZIP}}(y|x)$ as the probability mass function of a well-fitted Zero-Inflated Poisson model.

Define:

$$\hat{Y}(x) = E[Y|X=x] = \sum_{y=0}^{\infty} y \cdot f_{\text{ZIP}}(y|x)$$

$$\mathcal{L}_{\text{ZIP}}(f_{\text{ZIP}}, \mathcal{D}) = -\sum_{(x,y)\in\mathcal{D}} \log f_{\text{ZIP}}(y|x)$$

$$\mathcal{L}_{\text{rank}}(\hat{Y}, \mathcal{D}) = \sum_{(x_i,y_i),(x_j,y_j)\in\mathcal{D}:y_i>y_j} \max(0, \hat{Y}(x_j) - \hat{Y}(x_i) + \delta)$$

where $\mathcal{D}$ is the observed dataset and $\delta > 0$.

We aim to prove $I(\mathcal{L}_{\text{rank}}|\mathcal{L}_{\text{ZIP}}) > 0$, where $I(\cdot|\cdot)$ denotes conditional mutual information.

Consider $(x_i, y_i), (x_j, y_j) \in \mathcal{D}$ with $y_i > y_j$. It's possible that $\hat{Y}(x_i) \leq \hat{Y}(x_j)$ due to the nature of likelihood optimization in the ZIP model.

In this case:

$$\mathcal{L}_{\text{ZIP}}(f_{\text{ZIP}}, \{(x_i,y_i),(x_j,y_j)\}) = -\log f_{\text{ZIP}}(y_i|x_i) - \log f_{\text{ZIP}}(y_j|x_j)$$

$$\mathcal{L}_{\text{rank}}(\hat{Y}, \{(x_i,y_i),(x_j,y_j)\}) = \max(0, \hat{Y}(x_j) - \hat{Y}(x_i) + \delta) > 0$$

This implies:

$$P(Y_i > Y_j|\mathcal{L}_{\text{ZIP}}, \mathcal{L}_{\text{rank}}) > P(Y_i > Y_j|\mathcal{L}_{\text{ZIP}})$$

Consequently:

$$H(Y|\mathcal{L}_{\text{ZIP}}, \mathcal{L}_{\text{rank}}) < H(Y|\mathcal{L}_{\text{ZIP}})$$

Therefore, $I(\mathcal{L}_{\text{rank}}|\mathcal{L}_{\text{ZIP}}) = H(Y|\mathcal{L}_{\text{ZIP}}) - H(Y|\mathcal{L}_{\text{ZIP}}, \mathcal{L}_{\text{rank}}) > 0$. □

*Proof.* **[Proof of Theorem2]** Given Lemma1, We firstly prove that there exists a set of predictions $\hat{r}^C$ and a sufficiently small $\gamma_0 > 0$ such that for all $\gamma \in (0, \gamma_0)$:

$$E[\mathcal{L}_{\text{ZIP}}(\hat{r}^C, R)] - E[\mathcal{L}_{\text{ZIP}}(\hat{r}^{ZI}, R)] < \frac{1-\gamma}{\gamma}(E[\mathcal{L}_{\text{rank}}(\hat{r}^{ZI}, R)] - E[\mathcal{L}_{\text{rank}}(\hat{r}^C, R)])$$

Define the combined loss function $L_C(\hat{y}, Y; \alpha) = \alpha \mathcal{L}_{\text{ZIP}}(\hat{y}, Y) + (1-\alpha)\mathcal{L}_{\text{rank}}(\hat{y}, Y)$, where $\alpha \in (0, 1)$. Let $\hat{y}^C(\alpha)$ be the minimizer of $L_C$:

$$\hat{y}^C(\alpha) = \arg\min_{\hat{y}} E[L_C(\hat{y}, Y; \alpha)]$$

By the definition of $\hat{y}^C(\alpha)$, for any $\alpha \in (0, 1)$, we have:

$$E[L_C(\hat{y}^C(\alpha), Y; \alpha)] \leq E[L_C(\hat{y}^{\text{ZIP}}, Y; \alpha)]$$

Expanding this inequality:

$$\alpha E[\mathcal{L}_{\text{ZIP}}(\hat{y}^C(\alpha), Y)] + (1-\alpha)E[\mathcal{L}_{\text{rank}}(\hat{y}^C(\alpha), Y)] \leq \alpha E[\mathcal{L}_{\text{ZIP}}(\hat{y}^{\text{ZIP}}, Y)] + (1-\alpha)E[\mathcal{L}_{\text{rank}}(\hat{y}^{\text{ZIP}}, Y)]$$

Let $\Delta\mathcal{L}_{\text{ZIP}}(\alpha) = E[\mathcal{L}_{\text{ZIP}}(\hat{y}^C(\alpha), Y)] - E[\mathcal{L}_{\text{ZIP}}(\hat{y}^{\text{ZIP}}, Y)]$ and $\Delta\mathcal{L}_{\text{rank}}(\alpha) = E[\mathcal{L}_{\text{rank}}(\hat{y}^{\text{ZIP}}, Y)] - E[\mathcal{L}_{\text{rank}}(\hat{y}^C(\alpha), Y)]$. Rearranging the inequality:

$$\alpha \Delta \mathcal{L}_{\text{ZIP}}(\alpha) \leq (1 - \alpha) \Delta \mathcal{L}_{\text{rank}}(\alpha)$$

Given that $I(\mathcal{L}_{\text{rank}} | \mathcal{L}_{\text{ZIP}}) > 0$, $\mathcal{L}_{\text{rank}}$ provides information not captured by $\mathcal{L}_{\text{ZIP}}$. This implies that there exists $\alpha_1 \in (0, 1)$ such that for all $\alpha \in (0, \alpha_1]$, $\Delta \mathcal{L}_{\text{rank}}(\alpha) > 0$.

Now, consider the function:

$$f(\alpha) = (1 - \alpha) \Delta \mathcal{L}_{\text{rank}}(\alpha) - \alpha \Delta \mathcal{L}_{\text{ZIP}}(\alpha)$$

We know that $f(\alpha) \geq 0$ for all $\alpha \in (0, 1)$ from the earlier inequality. Moreover, $f(0) = \Delta \mathcal{L}_{\text{rank}}(0) > 0$ due to the information gain assumption.

By the continuity of $f(\alpha)$, there exists $\alpha_0 \in (0, \alpha_1]$ such that for all $\alpha \in (0, \alpha_0]$:

$$f(\alpha) > 0$$

This implies:

$$(1 - \alpha) \Delta \mathcal{L}_{\text{rank}}(\alpha) > \alpha \Delta \mathcal{L}_{\text{ZIP}}(\alpha)$$

Dividing both sides by $\alpha(1 - \alpha)$ (which is positive for $\alpha \in (0, 1)$):

$$\frac{\Delta \mathcal{L}_{\text{rank}}(\alpha)}{\alpha} > \frac{\Delta \mathcal{L}_{\text{ZIP}}(\alpha)}{1 - \alpha}$$

This is equivalent to:

$$\Delta \mathcal{L}_{\text{ZIP}}(\alpha) < \frac{1 - \alpha}{\alpha} \Delta \mathcal{L}_{\text{rank}}(\alpha)$$

Substituting back the definitions of $\Delta \mathcal{L}_{\text{ZIP}}(\alpha)$ and $\Delta \mathcal{L}_{\text{rank}}(\alpha)$:

$$E[\mathcal{L}_{\text{ZIP}}(\hat{y}^C(\alpha), Y)] - E[\mathcal{L}_{\text{ZIP}}(\hat{y}^{\text{ZIP}}, Y)] < \frac{1 - \alpha}{\alpha}(E[\mathcal{L}_{\text{rank}}(\hat{y}^{\text{ZIP}}, Y)] - E[\mathcal{L}_{\text{rank}}(\hat{y}^C(\alpha), Y)])$$

Let $\hat{y}^C = \hat{y}^C(\alpha_0)$, we have:

$$E[\mathcal{L}_{\text{ZIP}}(\hat{y}^C, Y)] - E[\mathcal{L}_{\text{ZIP}}(\hat{y}^{\text{ZIP}}, Y)] < \frac{1 - \alpha}{\alpha}(E[\mathcal{L}_{\text{rank}}(\hat{y}^{\text{ZIP}}, Y)] - E[\mathcal{L}_{\text{rank}}(\hat{y}^C, Y)])$$

Rearranging this inequality:

$$\alpha E[\mathcal{L}_{\text{ZIP}}(\hat{y}^C, Y)] + (1 - \alpha) E[\mathcal{L}_{\text{rank}}(\hat{y}^C, Y)] < \alpha E[\mathcal{L}_{\text{ZIP}}(\hat{y}^{\text{ZIP}}, Y)] + (1 - \alpha) E[\mathcal{L}_{\text{rank}}(\hat{y}^{\text{ZIP}}, Y)]$$

The left-hand side of this inequality is $E[L_C(\hat{y}^C)]$ by definition. The right-hand side is strictly greater than $E[\mathcal{L}_{\text{ZIP}}(\hat{y}^{\text{ZIP}})]$ since $E[\mathcal{L}_{\text{rank}}(\hat{y}^{\text{ZIP}}, Y)] > 0$ for any non-trivial ranking loss and $\alpha < 1$.

Therefore:

$$E[L_C(\hat{y}^C)] < \alpha E[\mathcal{L}_{\text{ZIP}}(\hat{y}^{\text{ZIP}}, Y)] + (1 - \alpha) E[\mathcal{L}_{\text{rank}}(\hat{y}^{\text{ZIP}}, Y)] < E[\mathcal{L}_{\text{ZIP}}(\hat{y}^{\text{ZIP}})]$$

This completes the proof. $\qquad\square$

## A.2 GRADIENT ANALYSIS

We analyze the composite ranking loss function $\mathcal{L}_{\text{rank}}$, which combines Pairwise Soft Ranking Loss and Listwise Global Ranking Loss. The gradient of $\mathcal{L}_{\text{rank}}$ with respect to $\hat{y}_i$ is:

$$\frac{\partial \mathcal{L}_{\text{rank}}}{\partial \hat{y}_i} = \beta \frac{\partial \mathcal{L}_{\text{PSR}}}{\partial \hat{y}_i} + (1 - \beta) \frac{\partial \mathcal{L}_{\text{LGR}}}{\partial \hat{y}_i} \tag{12}$$

$$\frac{\partial \mathcal{L}_{\text{PSR}}}{\partial \hat{y}_i} = -\left( \sum_{j \neq i}(\Delta_{ij} \cdot \sigma_{ij}) - \sum_{j \neq i}(\Delta_{ji} \cdot \sigma_{ji}) \right) - \hat{y}_i \sum_{j \neq i} \Delta_{ij} \cdot \frac{\partial \sigma_{ij}}{\partial \hat{y}_i} + \hat{y}_i \sum_{j \neq i} \Delta_{ji} \cdot \frac{\partial \sigma_{ji}}{\partial \hat{y}_i} \tag{13}$$

where

$$\frac{\partial \sigma_{ij}}{\partial \hat{y}_i} = \frac{\text{sign}(\hat{y}_i - \hat{y}_j)}{T} \sigma_{ij}(1 - \sigma_{ij}) \tag{14}$$

The gradient $\frac{\partial \mathcal{L}_{\text{PSR}}}{\partial \hat{y}_i}$ is primarily determined by $\Delta_{ij}$ and $\sigma_{ij}$, which represent pairwise comparisons between item $i$ and other items $j$. $\Delta_{ij}$ captures the NDCG impact of swapping items $i$ and $j$, while $\sigma_{ij}$ adjusts this impact based on the difference between $\hat{y}_i$ and $\hat{y}_j$. This formulation ensures that $\mathcal{L}_{\text{PSR}}$ focuses on local ranking relationships, particularly between adjacent or nearby items.

$$\frac{\partial \mathcal{L}_{\text{LGR}}}{\partial \hat{y}_i} = -\frac{1}{T} \sum_{k=i}^{n} \left( \frac{\exp(\hat{y}_{\pi(k)}/T)}{\sum_{j=k}^{n} \exp(\hat{y}_{\pi(j)}/T)} - \mathbb{1}[\pi(k) = i] \right) + \frac{\partial \mathcal{L}_{\text{con}}}{\partial \hat{y}_i} \tag{15}$$

The gradient $\frac{\partial \mathcal{L}_{\text{LGR}}}{\partial \hat{y}_i}$ incorporates information from all items ranked from position $i$ to $n$. Through its softmax formulation, it considers the position of item $i$ relative to all items ranked below it. This allows $\mathcal{L}_{\text{LGR}}$ to capture global ranking information.

## B Experimental Settings

### B.1 Virtual Docking for Dataset Construction

We employed molecular docking to define the three-dimensional conformations of molecules within our DEL datasets. This method was applied to both the training and evaluation sets, generating lig- and binding poses for all molecules. We concentrated on three pivotal carbonic anhydrase proteins: Q16790 (CAH9_HUMAN), P00918 (CAH2_HUMAN), and O43570 (CAH12_HUMAN).

For the Q16790 target, we sourced the 5fl4 and 2hkf PDB structures from the PDBbind database and utilized the Gerry dataset (Gerry et al., 2019). which comprised 108,529 molecules, generating up to nine potential poses per molecule. For the targets P00918 and O43570, we selected 127,500 SMILES strings from the DEL-MAP dataset (Hou et al., 2023) and conducted self-docking using the 3p3h and 5doh PDB structures for P00918, and 4kp5 and 4ht2 for O43570, as sourced from PDBbind. For the validation set, we applied the same docking methodology to the corresponding ligands of CA9, CA2, and CA12, involving 3,324, 6,395, and 2,690 ligands respectively.

In the specific docking procedures, initial 3D conformations of ligands were created using RDKit. The binding sites in the protein-ligand complexes were identified using 3D structural data of known binding ligands from PDBbind as reference points. Targeted docking was performed by defining the search space as a 22.5 Å cube centered on the reference ligand in the corresponding PDBbind complex. Using SMINA docking software, we generated 9 potential poses for each protein-ligand pair.

### B.2 Hyperparameter Setting

The model was trained using the Adam optimizer with mini-batches of 64 samples. The network architecture employed a hidden dimension of 128. The self-correction mechanism was applied for 3 iterations. All experiments were conducted on a single NVIDIA RTX 3090 GPU with 24GB memory. The implementation utilized PyTorch-Lightning version 1.9.0 to streamline the training process and enhance reproducibility. The hyperparameter settings for different datasets, including loss function weights, temperature, and margin, are detailed in Table B.2.

**Hyper-parameter Selection** The hyperparameter configuration in the appendix requires clarification regarding the weight settings. The key parameters include $\mathcal{L}_{rank}$ weight, $\mathcal{L}_{PSR}$ weight, $\mathcal{L}_{LGR}$ weight, and ARC weight. The $\mathcal{L}_{rank}$ weight is logarithmically distributed between 1e9 and 1e11 to align with the magnitude of ZIP loss. $\mathcal{L}_{PSR}$ and $\mathcal{L}_{LGR}$ weights are calibrated to maintain appropriate balance among different ranking objectives. Given that ARC loss naturally aligns with ZIP loss magnitude, its weight is simply set to 1.0 or 0.1.

Temperature settings are determined by the characteristics of DEL read count data distribution, with denser distributions requiring lower temperatures. A detailed analysis of read count distribution and supporting theoretical proposition are provided in the Section C.1. Besides, the contract weight and margin serve as penalty terms for $\mathcal{L}_{LGR}$, with the weight selected based on $\mathcal{L}_{LGR}$'s relative magnitude. Detailed in Table B.2, these values remain stable and consistent across experiments.

Table 5: Hyperparameter Settings for DEL-Ranking on Different Datasets

|  | 3p3h | 4kp5-A | 4kp5-OA | 5fl4-9p | 5fl4-20p |
|---|---|---|---|---|---|
| $\mathcal{L}_{consist}$ weight $\gamma$ | 1 | 0.1 | 0.1 | – | – |
| $\mathcal{L}_{rank}$ weight $\rho$ | 1e11 | 1e9 | 1e10 | 1e8 | 1e8 |
| $\mathcal{L}_{PSR}$ weight $\beta$ | 0.5 | 0.91 | 0.91 | 0.67 | 0.5 |
| $\mathcal{L}_{LGR}$ weight 1-$\beta$ | 0.5 | 0.09 | 0.09 | 0.33 | 0.5 |
| Temperature $T$ | 0.8 | 0.3 | 0.2 | 0.9 | 0.2 |
| $\mathcal{L}_{con}$ weight $\sigma$ | 1e−3 | 1e−3 | 1e−3 | 1e−4 | 1e−3 |
| Margin $\tau$ | 1e−3 | 1e−3 | 1e−3 | 1e−3 | 1e−3 |

**Proposition 1.** *As $T \to 0$, the model simultaneously achieves: (1) **Near-deterministic selection** of compounds with the highest read counts, corresponding to the highest binding affinities; (2) **Increased robustness** to small noise perturbations in the DEL experiment data.*

Based on the Proposition, the adaptive-ranking model would obtain more consistent identification of high-affinity compounds, reducing errors due to random fluctuations. Also, it achieves enhanced robustness against common DEL experimental noises such as PCR bias and sequencing errors. While lowering the temperature leads to a more deterministic ranking with high-affinity sensitivity and noise resistance, there exists overlooking of compounds with slightly lower rankings when the temperature goes to extremely low. In experiments, we demonstrate that [0.1, 0.4] should be a proper range for the distribution sharping.

## C EXPERIMENTAL RESULTS

### C.1 ABLATION STUDY ON HYPER-PARAMETERS

In order to evaluate the robustness of our method, we conduct a comprehensive analysis of four critical hyperparameters: the consistency loss weight $\gamma$, ranking loss weight $\rho$, LGR loss weight $\beta$, and temperature $T$ across three datasets (3p3h, 4kp5-A, and 4kp5-OA). As shown in Table 6, we employ logarithmic search spaces for all loss-related hyperparameters to align the magnitudes of ranking and consistency losses with the ZIP loss, while adopting a linear search space for temperature.

The empirical results demonstrate that our selected hyperparameters consistently achieve optimal performance across all search spaces. The model exhibits strong stability, with performance variations remaining minimal under most hyperparameter adjustments. Nevertheless, we observe dataset-specific sensitivities: the 4kp5-OA dataset shows increased sensitivity to ranking loss weight variations, potentially due to elevated read count noise levels. Similarly, the 4kp5-A dataset exhibits performance fluctuations at higher values of ranking loss and $\mathcal{L}_{LGR}$ weights, which we attribute to magnitude imbalances in the numerical representations.

The performance progression with respect to temperature demonstrates a consistent linear relationship, providing empirical support for our distribution sharpening hypothesis. These findings collectively indicate that while our model maintains robustness across the hyperparameter search space with well-justified parameter selections, its sensitivity can be influenced by dataset-specific characteristics, particularly read count distribution noise and magnitude disparities in the underlying data.

Table 6: Comparison of different hyper-parameters on binding affinity prediction performance. The best performance within one set of hyperparameter group is set **bold**.

| Parameter | Value | 3p3h (CA2) | | 4kp5-A (CA12) | | 4kp5-OA (CA12) | |
|---|---|---|---|---|---|---|---|
| Metric | | Sp | SubSp | Sp | SubSp | Sp | SubSp |
| $\mathcal{L}_{consist}$ weight $\gamma$ | 0.1 | $-0.275 \pm 0.011$ | $-0.163 \pm 0.017$ | $\mathbf{-0.268 \pm 0.012}$ | $\mathbf{-0.277 \pm 0.016}$ | $\mathbf{-0.289 \pm 0.025}$ | $\mathbf{-0.233 \pm 0.021}$ |
| | 1 | $\mathbf{-0.286 \pm 0.002}$ | $\mathbf{-0.177 \pm 0.005}$ | $-0.266 \pm 0.008$ | $-0.238 \pm 0.008$ | $-0.287 \pm 0.005$ | $-0.213 \pm 0.014$ |
| | 10 | $-0.276 \pm 0.010$ | $-0.163 \pm 0.015$ | $-0.258 \pm 0.019$ | $-0.239 \pm 0.010$ | $-0.278 \pm 0.024$ | $-0.227 \pm 0.040$ |
| $\mathcal{L}_{rank}$ weight $\rho$ | 1e9 | $-0.266 \pm 0.011$ | $-0.151 \pm 0.016$ | $\mathbf{-0.268 \pm 0.012}$ | $\mathbf{-0.277 \pm 0.016}$ | $-0.152 \pm 0.045$ | $-0.225 \pm 0.023$ |
| | 1e10 | $-0.269 \pm 0.006$ | $-0.151 \pm 0.009$ | $-0.257 \pm 0.005$ | $-0.189 \pm 0.016$ | $\mathbf{-0.289 \pm 0.025}$ | $\mathbf{-0.233 \pm 0.021}$ |
| | 1e11 | $\mathbf{-0.286 \pm 0.002}$ | $\mathbf{-0.177 \pm 0.005}$ | $-0.135 \pm 0.012$ | $-0.060 \pm 0.036$ | $-0.084 \pm 0.095$ | $-0.058 \pm 0.077$ |
| $\mathcal{L}_{LGR}$ weight $\beta$ | 0.09 | $-0.277 \pm 0.009$ | $-0.165 \pm 0.013$ | $\mathbf{-0.268 \pm 0.012}$ | $\mathbf{-0.277 \pm 0.016}$ | $\mathbf{-0.289 \pm 0.025}$ | $\mathbf{-0.233 \pm 0.021}$ |
| | 0.5 | $\mathbf{-0.286 \pm 0.002}$ | $\mathbf{-0.177 \pm 0.005}$ | $-0.267 \pm 0.033$ | $-0.240 \pm 0.016$ | $-0.288 \pm 0.025$ | $-0.247 \pm 0.019$ |
| | 0.91 | $-0.275 \pm 0.011$ | $-0.160 \pm 0.019$ | $-0.173 \pm 0.054$ | $-0.089 \pm 0.038$ | $-0.279 \pm 0.007$ | $-0.222 \pm 0.033$ |
| Temperature $T$ | 0.2 | $-0.280 \pm 0.021$ | $-0.173 \pm 0.029$ | $-0.267 \pm 0.013$ | $-0.247 \pm 0.009$ | $\mathbf{-0.289 \pm 0.025}$ | $\mathbf{-0.233 \pm 0.021}$ |
| | 0.5 | $-0.279 \pm 0.009$ | $-0.169 \pm 0.014$ | $-0.266 \pm 0.014$ | $-0.236 \pm 0.012$ | $-0.275 \pm 0.013$ | $-0.216 \pm 0.005$ |
| | 0.8 | $\mathbf{-0.286 \pm 0.002}$ | $\mathbf{-0.177 \pm 0.005}$ | $-0.268 \pm 0.010$ | $-0.222 \pm 0.010$ | $-0.275 \pm 0.035$ | $-0.220 \pm 0.029$ |
| | 1 | $-0.279 \pm 0.011$ | $-0.166 \pm 0.015$ | $-0.247 \pm 0.022$ | $-0.265 \pm 0.014$ | $-0.256 \pm 0.033$ | $-0.181 \pm 0.046$ |

### C.2 ABLATION STUDY ON STRUCTURE INFORMATION

To assess the value of structural information from docking software and its complementarity with sequence features, we performed an ablation study focusing on the additive combination of structure and fingerprint embeddings in the ARC algorithm. We applied varying scaling factors (0, 0.3, 0.6, 1.0, 1.5, and 2.0) to the structure embedding across three datasets (3p3h, 4kp5-A, and 4kp5-OA) with five random seeds. Table 7 shows that incorporating structural information significantly improves model performance. The analysis revealed higher model sensitivity in the noise-prone 4kp5-OA dataset, while performance degradation was observed in 4kp5-A when scaling factors exceeded 1.0. These results indicate that while structural information enhances model performance, excessive weighting of potentially uncertain structural data can impair predictions. Nevertheless, our chosen parameterization demonstrates consistent performance across all datasets.

### C.3 COMPARISON RESULT OF TOP-50 SELECTION CASES BY DEL-DOCK

Table 7: Parameter value comparison for structure scaling factor. The best performance within one set of hyperparameter group is set **bold**.

| Value $\varsigma$ | 3p3h (CA2) | | 4kp5-A (CA12) | | 4kp5-OA (CA12) | |
|---|---|---|---|---|---|---|
| Metric | Sp | SubSp | Sp | SubSp | Sp | SubSp |
| 0 | $-0.236\pm_{0.010}$ | $-0.112\pm_{0.013}$ | $-0.253\pm_{0.012}$ | $-0.218\pm_{0.017}$ | $-0.195\pm_{0.044}$ | $-0.103\pm_{0.055}$ |
| 0.3 | $-0.262\pm_{0.008}$ | $-0.145\pm_{0.012}$ | $-0.265\pm_{0.017}$ | $-0.227\pm_{0.017}$ | $-0.124\pm_{0.146}$ | $-0.062\pm_{0.090}$ |
| 0.6 | $-0.263\pm_{0.008}$ | $-0.146\pm_{0.011}$ | $-0.250\pm_{0.017}$ | $-0.231\pm_{0.019}$ | $-0.210\pm_{0.040}$ | $-0.121\pm_{0.047}$ |
| 1 | $\mathbf{-0.286\pm_{0.002}}$ | $\mathbf{-0.177\pm_{0.005}}$ | $\mathbf{-0.268\pm_{0.012}}$ | $\mathbf{-0.277\pm_{0.016}}$ | $\mathbf{-0.289\pm_{0.025}}$ | $\mathbf{-0.233\pm_{0.021}}$ |
| 1.5 | $-0.270\pm_{0.011}$ | $-0.155\pm_{0.016}$ | $-0.244\pm_{0.022}$ | $-0.252\pm_{0.022}$ | $-0.152\pm_{0.156}$ | $-0.139\pm_{0.104}$ |
| 2 | $-0.271\pm_{0.012}$ | $-0.155\pm_{0.015}$ | $-0.191\pm_{0.089}$ | $-0.216\pm_{0.060}$ | $-0.230\pm_{0.038}$ | $-0.152\pm_{0.051}$ |

To evaluate the DEL-Dock model's performance, we analyzed the $K_i$ value distributions and benzenesulfonamide identification rates for the top-50 compounds across multiple datasets. Figure 4 demonstrates that DEL-Dock outperformed DEL-Ranking in the 3p3h dataset and showed comparable results in 4kp5-A and 5fl4-9p datasets, while DEL-Ranking exhibited superior performance in 4kp5-OA and 5fl4-20p datasets. The benzenesulfonamide identification accuracy mirrored these trends across all datasets, with DEL-Dock showing strength in 3p3h, equivalent performance in 4kp5-A and 5fl4-9p, and DEL-Ranking maintaining advan-

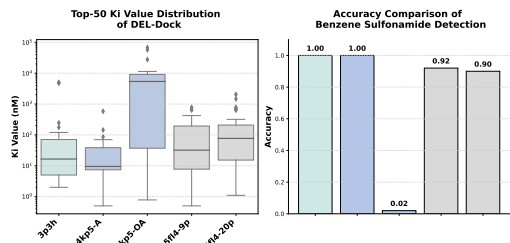

Figure 4: Quantitative analysis of Top-50 selection, including $K_i$ distribution and accuracy for DEL-Dock (Shmilovich et al., 2023).

tage in 4kp5-OA and 5fl4-20p datasets. Further analysis indicated that ranking-based methods performed better in datasets with higher noise levels and increased read counts, aligning with theoretical expectations.

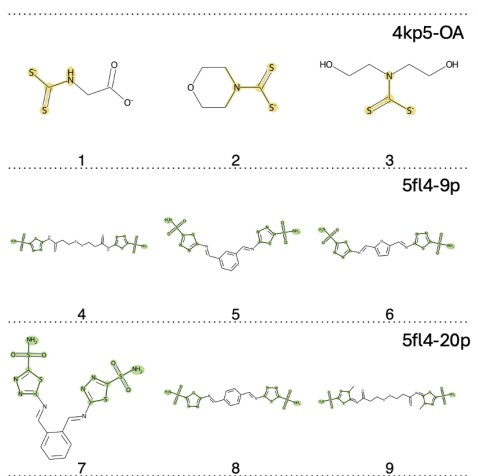

Figure 5: Visualization of Top-50 high affinity cases without benzene sulfonamide.

Moreover, analysis of high-affinity compounds devoid of benzenesulfonamide functional groups was performed across the 4kp5-OA, 5fl4-9p, and 5fl4-20p datasets. The identification of thiocarbonyl and sulfonamide groups exhibiting Ki values below 10.0 yielded two significant insights. First, it validated the benzenesulfonamide-based activity labeling approach, particularly given that DEL-Dock achieved these results independent of activity label data. Second, the label-guided DEL-Ranking model successfully identified structurally analogous functional groups to benzenesulfonamides, demonstrating that ranking supervision effectively enhances the discovery of novel, high-activity molecular scaffolds.

## C.4 VISUALIZATION OF TOP-50 SELECTION OF DEL-RANKING

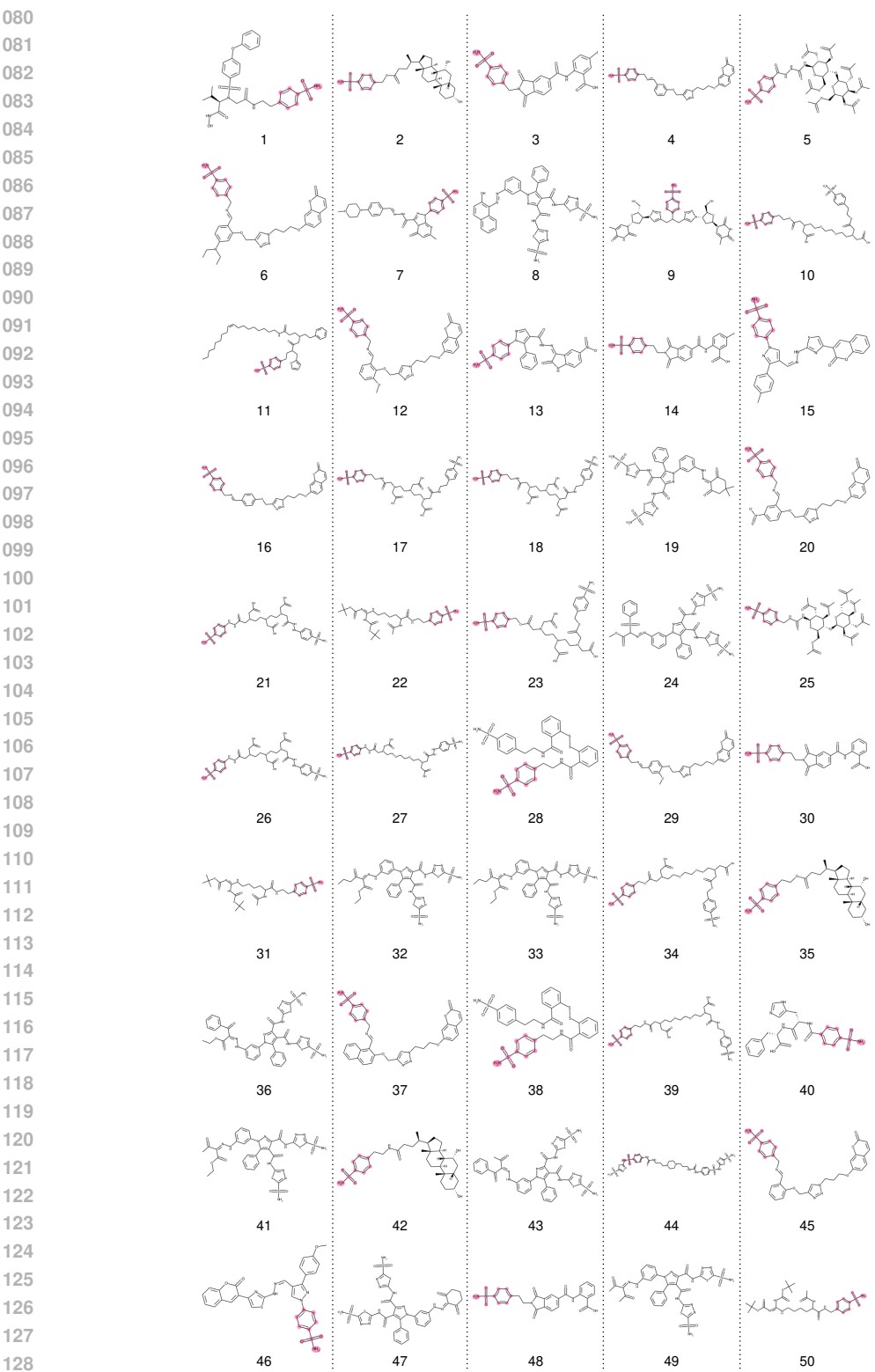

Figure 6: Visualization of the top-50 DEL-Ranking results on the 3p3h dataset. In molecules containing benzenesulfonamide, the benzenesulfonamide structure is highlighted.

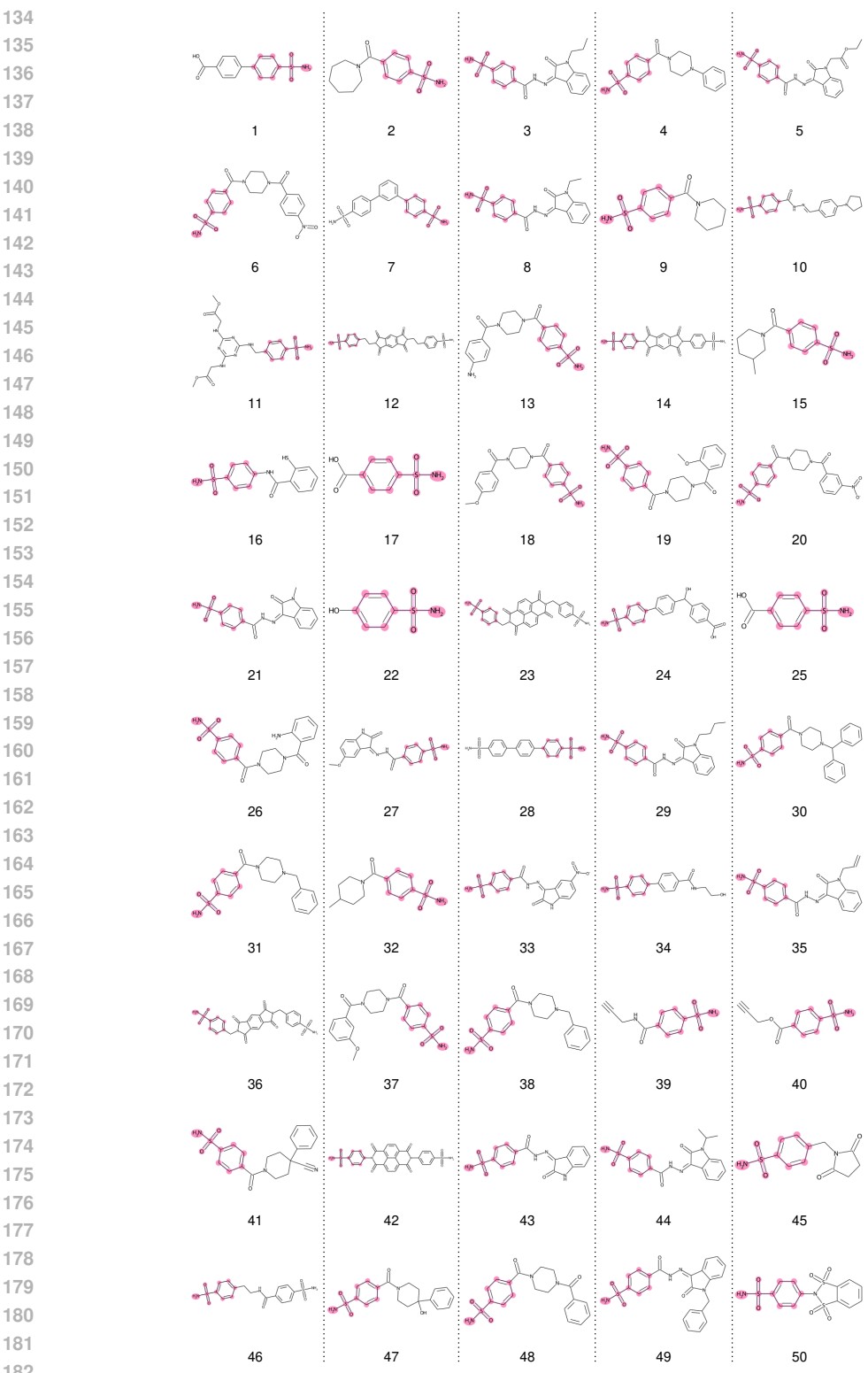

Figure 7: Visualization of the top-50 DEL-Ranking results on the 4kp5-A dataset. In molecules containing benzenesulfonamide, the benzenesulfonamide structure is highlighted.

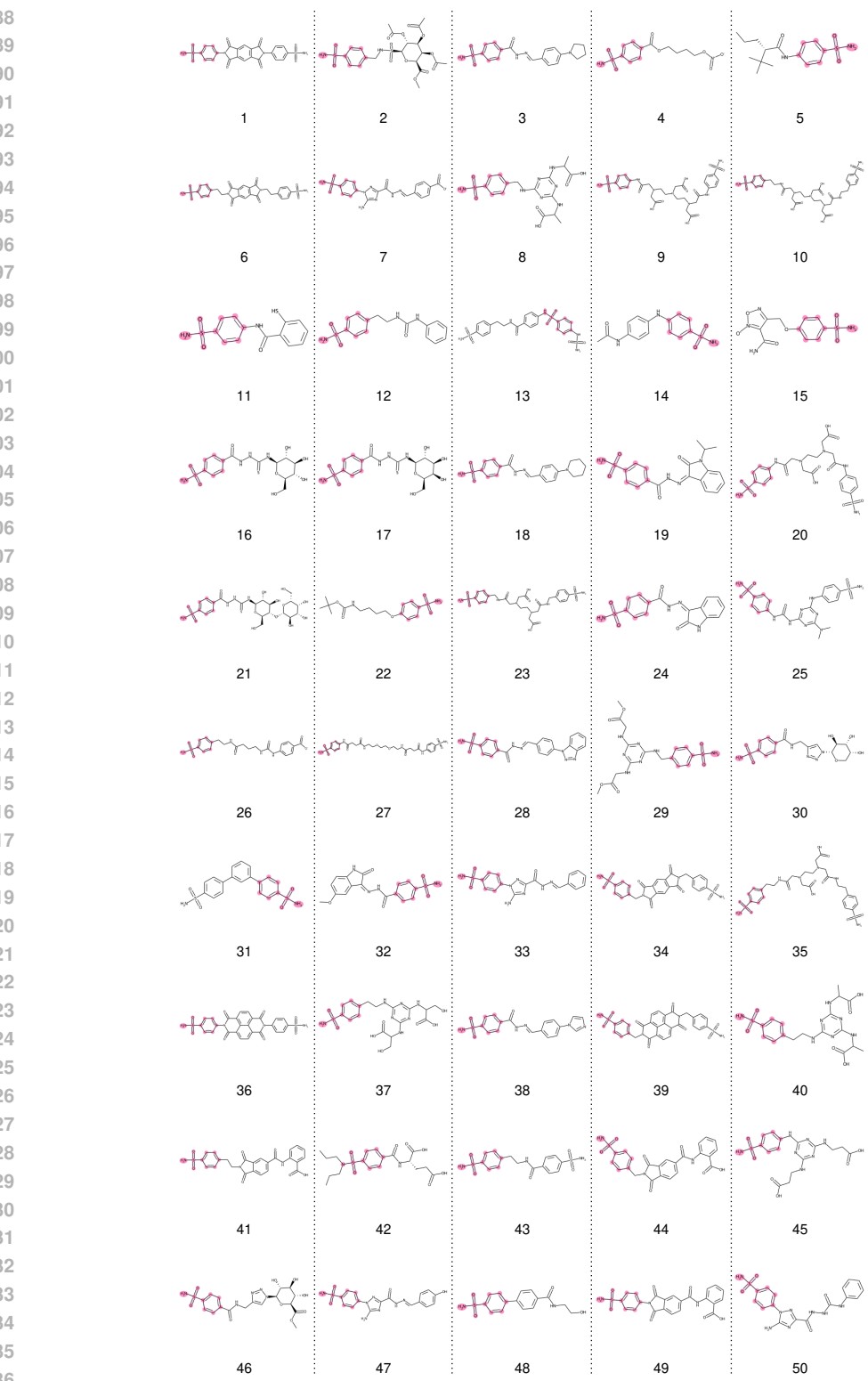

Figure 8: Visualization of the top-50 DEL-Ranking results on the 4kp5-OA dataset. In molecules containing benzenesulfonamide, the benzenesulfonamide structure is highlighted.

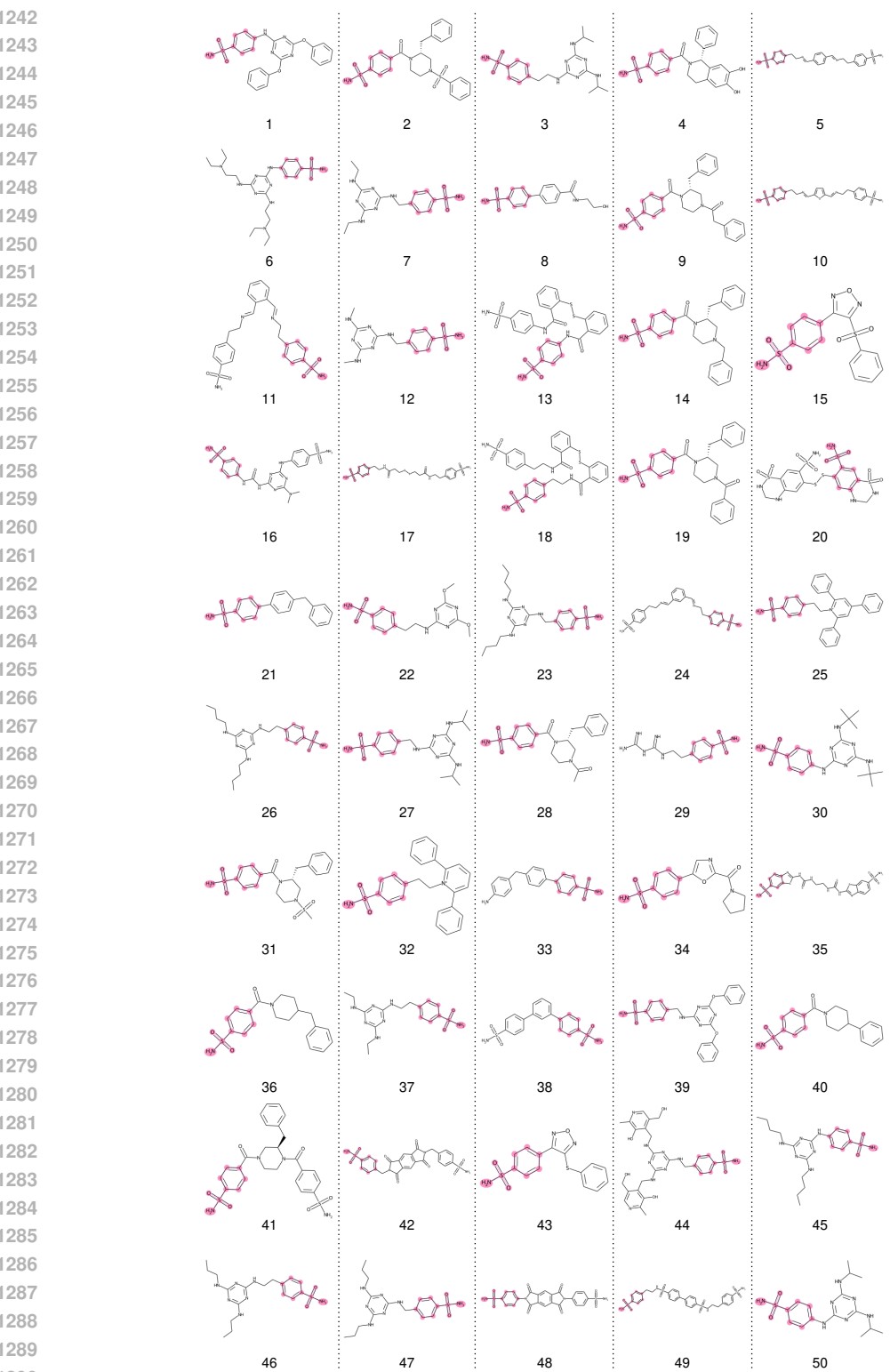

Figure 9: Visualization of the top-50 DEL-Ranking results on the 5fl4(9 pose) dataset. In molecules containing benzenesulfonamide, the benzenesulfonamide structure is highlighted.

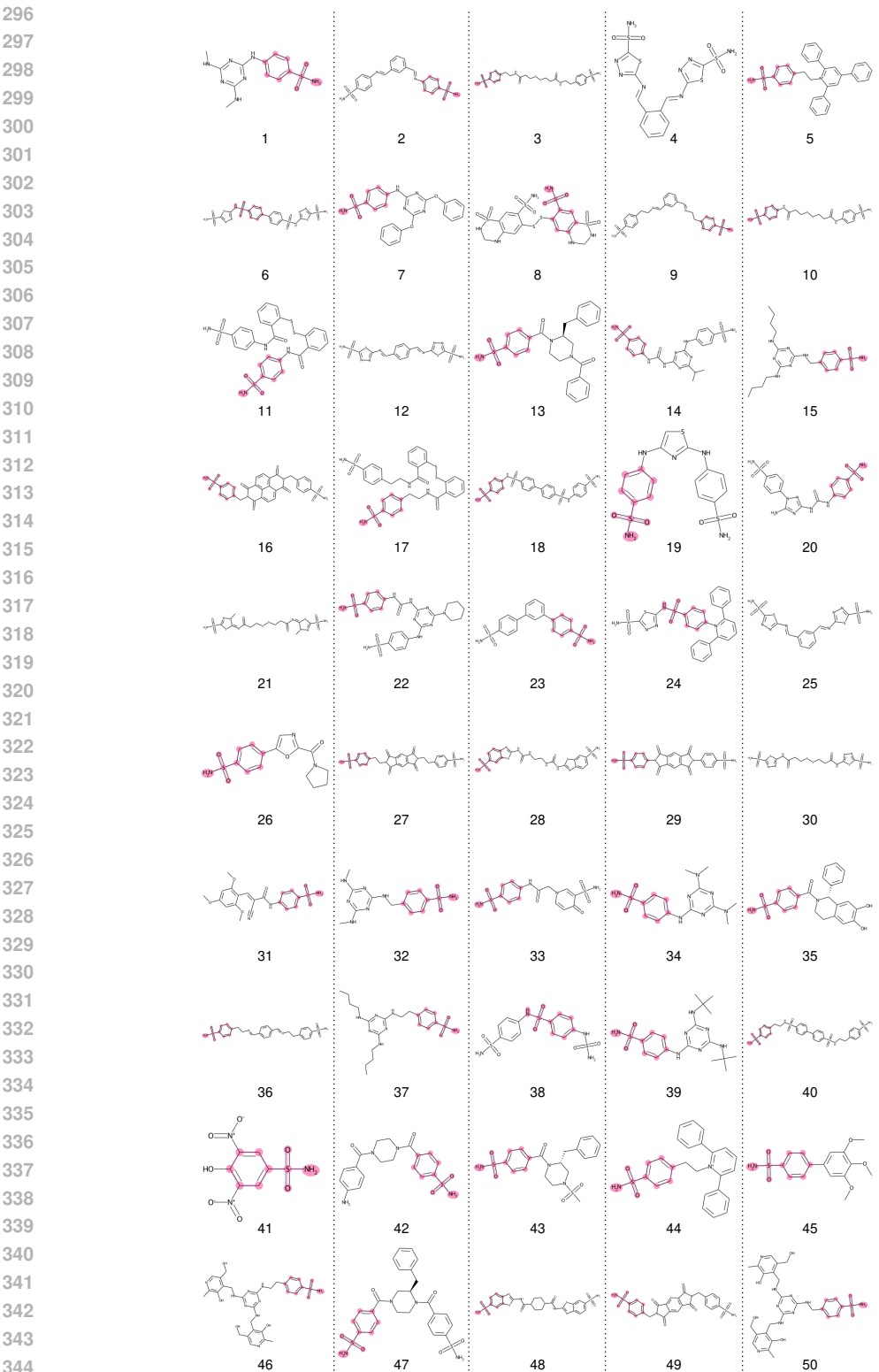

Figure 10: Visualization of the top-50 DEL-Ranking results on the 5fl4(20 pose) dataset. In molecules containing benzenesulfonamide, the benzenesulfonamide structure is highlighted.

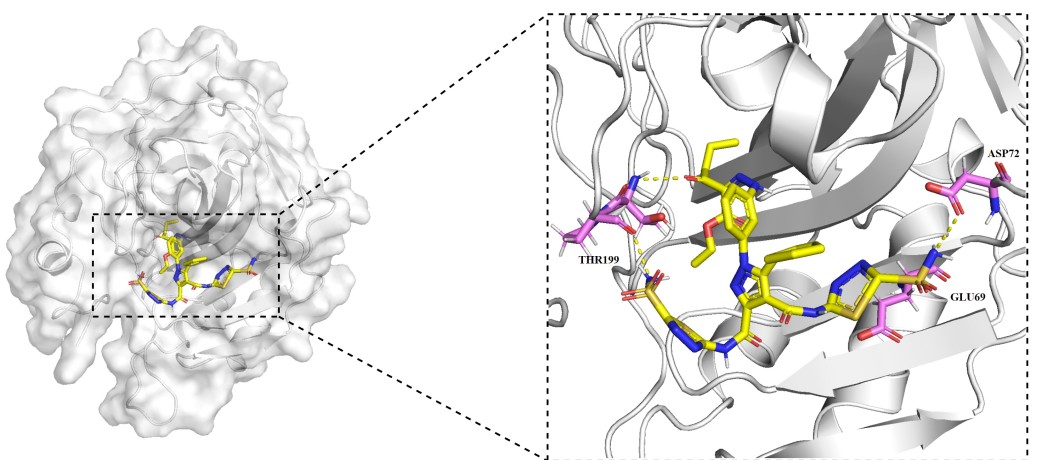

Figure 11: In 3p3h, THR199 likely forms hydrogen bonds with the ligand, while ASP72 and GLU69 participate in hydrogen bonding and electrostatic interactions. The corresponding $k_i$ value is 84.0.

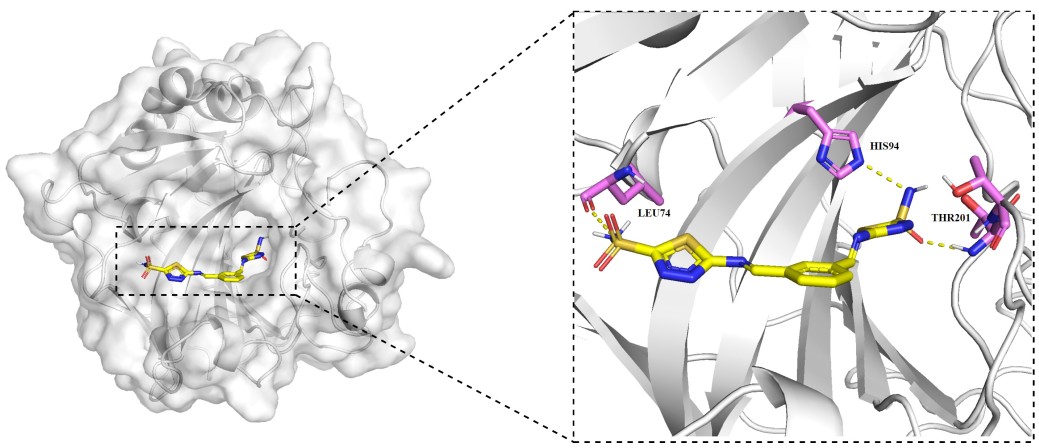

Figure 12: In 5fl4, LEU74 contributes through van der Waals forces or hydrophobic interactions, HIS94's imidazole side chain potentially forms hydrogen bonds, and THR201 engage in hydrogen bonding with the ligand. The corresponding $k_i$ value is 0.5.

