# OpenReview forum: "DEL-Ranking: Ranking-Correction Denoising Framework for Elucidating Molecular Affinities in DNA-Encoded Libraries"
_ICLR.cc/2025/Conference — Submitted to ICLR 2025_

### Official Review · Reviewer_jJbZ · 2024-11-02

**Soundness:** 2
**Presentation:** 1
**Contribution:** 2
**Rating:** 3
**Confidence:** 3

**Summary:**

This paper tackles the problem of how to build a good machine learning model of ligand binding activity based on DEL screens. The two primary challenges with modeling DEL-based data is the noise in the data and the mismatch between the true log-enrichment of a compound in the DEL assay and the actual binding affinity of the compound. The authors propose a number of new modeling approaches to tackle these issues. First, a new ranking-loss is proposed. In addition the authors introduce an iterative training approach that promotes the matching of read-based predictions to true activity values as well as a consistency loss.

**Strengths:**

The paper addresses a number of relevant challenges to an important problem of predicting binding affinity from DEL data.

**Weaknesses:**

1) I found it really hard to understand the details of the specific losses and approaches. For example, Equation 4 seems to depict a function that takes in two element, y_i and y_j, but the right-hand side is written using a sum of 'i' and 'j', suggesting it is over the entire dataset?

I was unable to understand at all what the activity-guided refinement is based on the written text. Similarly, there is no clear motivation for the specific choices of many of the loss functions. For example, Equation 8 is just presented but there is no explanation for why the particular terms are included over a simpler loss such as ||A_true - A_pred||_2.

2) The largest weakness for me is that the paper seems reads as a "bag-of-tricks". Each individual loss may be reasonably motivated, but each loss term also introduces its own set of hyperparameters. This paper introduces at least 6 new hyperparameters. Importantly, there was no discription of how these hyperparameters were chosen and no demonstration that they can be robustly chosen using a validation set. Thus, I do not believe the authors have demonstrated that they have actually produced a more useful method. So far, they have only shown that with a carefully chosen set of hyperparameters and the introduction of a much much more complicated modeling scheme that they can slightly improve the spearman correlations.

3) Nowhere in this paper is the ZIP loss actually defined. It first shows up in the paper with just the acronym and no description of what ZIP means.

**Questions:**

1) How were the hyperparameters chosen?

2) Can they be chosen robustly with a validation set? Please provide experiments demonstrating this, if so.

3) Using molecular docking structures seems potentially problematic as these poses are often wrong and therefore add noise to your model. Please discuss this further.

4) The authors claim that DEL data is inherently ranking in nature, but this is not obvious to me. The relative magnitude of the log-enrichments may provide useful information about the binding affinity. Throwing away this information and only ranking loses information.

5) What is ZIP?

6) Please provide further discussion of the iterative algorithm.

---

> ### Author Response · Authors · 2024-11-23
> **Response to Reviewer jJbZ**
>
> We greatly appreciate your detailed review and insightful questions! We have addressed all points you raised in the "Weakness" and "Questions" sections. We organize your comments along with our response into the following three aspects:
>
> ## Article Presentation
>
> > I found it really hard to understand the details ... over a simpler loss such as ||A_true - A_pred||_2.
>
> We apologize for the lack of an overall introduction that motivates our proposed loss objectives to train a ML model for the DEL prediction task. For your convenience, we summarize here the motivations that we now put into the revised manuscript, Firstly, we denote ZIP as Zero-Inflated Poisson Distribution, and ZIP loss as the expectation of Negative Log-Likelihood (NLL) for all predicted read count values.
>
> a)	L_rank: we claim that fitting the ZIP distribution with ZIP loss can only obtain an unordered read-count distribution, where there is no supervision on mapping each compound to its read-count values (As we claimed in Section3.2). So, we would like to propose a ranking-based loss to help the model to understand the relative ranking and order of compounds to obtain a correct order. Our lemma and theorem have proven that ranking brings new information to the original system so as to lower the loss expectation.
>
> b)	Following the motivation behind L_rank, it is natural to design a ranking loss function with supervision signal of different views, which are pair-wise and list-wise. These two types of ranking loss can make up for each other’s shortcomings and make the ranking loss function more robust.
>
> c)	L_PSR: As stated in Section 3.2.1, we propose this pair-wise ranking loss inspired by LambdaRank [3], where there are relative gains and discount functions are proposed to capture the relative ranking changes. We improve this with two points: 1) We make it smooth with approximation of relative gain and discount as \Delta G and \Delta D to make the loss function smooth and differentiable. 2) We set top-K samples of one batch to improve the efficiency and stability based on the nature of DEL data, where there is a portion of zeros. Ranking based on zero pairs is inefficient and confuses the model’s ranking process.
>
> d) L_LGR: As stated in Section 3.2.2, we propose this list-wise ranking loss to enhance the ranking ability of L_PSR for a more robust ranking. The design of L_LGR is inspired by ListMLE [2], where a list-wise cross entropy loss is utilized to describe the global mismatch between two lists. However, traditional ListMLE loss contains shortcomings when used in DEL data: 1) The original DEL read-count values contain many zeros, which leads to the similar problem as L_PSR. So we propose L_con as the contrastive loss between any two read-count values to enlarge the gaps between their predictions. This contrastive loss could bring more information to the relative binding-affinity information of zero-read-count compounds so that we can potentially utilize more information for our ranking function. 2) The original data distribution contains to much read-count values that are close to each other caused by the inherent noise from DEL experiments. When using list-wise ranking functions, the compounds with close predicted read-count values cannot be distinguished well since the corresponding cross-entropy loss value is relatively small. So, here we propose the Temperature to make the distribution more sharpening and smoother. Smaller temperature leads to more sharpening distribution, which can benefit 1) Near-deterministic selection and 2) Increased robustness, as claimed in Section 3.2.2.
>
> e) ARC (Activity-Referenced Correction): This algorithm is proposed to utilize the information of the binary activity labels in our dataset, where the binary label is annotated as 1 if there is Benzene Sulfonamide in the compound, as suggested in [3]. So the activity label would bring more information based on compound structures, and simultaneously enhance the read-count value regression. For instance, consider two compounds with similar read-count values but with different binary labels, the model can distinguish them with this type of additional information, and further remove the distribution noise in read-count values. One thing we need to address is that the high affinity prior knowledge may not always exist in all protein targets. To be more complex, we need to design an efficient fusion algorithm to inject activity information to the read-count prediction. So, we propose the Refinement stage with iterative value injection to conduct multi-round update. However, here is a very common situation, like if there’s noise in read-count prediction or noise in activity prediction, how to minimize the side effect of large-error prediction? Then we enhance the L_consist with the second term in Section 3.3 to bound the prediction of read-count value and activity to avoid larger noise effect, which is the motivation of “particular terms” mentioned in the question.

---

> ### Author Response · Authors · 2024-11-23
> **Response to Reviewer jJbZ (continued)**
>
> > Nowhere in this paper is the ZIP loss actually defined. It first shows up in the paper with just the acronym and no description of what ZIP means.
>
> We have revised Section 3.1 to properly introduce and define Zero-Inflated Poisson (ZIP) distribution and its associated loss function. The ZIP loss is formulated as the negative log-likelihood expectation over both control counts and target counts:
>
> \begin{equation}
> \mathcal{L}_{\text{ZIP}}= -\sum_i \log[P(\hat{M}_i|\lambda_M, \pi_M)] -\sum_j \log[P(\hat{R}_j|\lambda_M + \lambda_R, \pi_R)]
> \end{equation}
>
> where:
> - $\lambda_M$, $\lambda_R$ represent the Poisson mean parameters for control and target counts respectively
> - $\pi_M$, $\pi_R$ denote the zero-excess probabilities for control and target counts respectively
>
>
> > What is ZIP?
>
> ZIP stands for Zero-Inflated Poisson distribution, a statistical model that accounts for excess zeros in count data. In our context, it models DEL read-count data through two components: a Poisson distribution for non-zero counts and a zero-inflation parameter to handle the excess zero observations. The complete mathematical formulation and its application to DEL data are now detailed in Section 3.1 of the revised manuscript.
>
>
> > Please provide further discussion of the iterative algorithm
>
> The Activity-Referenced Correction (ARC) algorithm iteratively refines predictions by leveraging both read-count and structural activity information. Each iteration consists of:
>
> 1. Read-count prediction refinement using current activity estimates
> 2. Activity prediction update incorporating refined read-count values
> 3. Consistency enforcement between predictions via bounded optimization
>
> The iterative nature allows gradual integration of structural activity information while preventing error accumulation through consistency constraints. We have expanded this explanation with mathematical details and convergence analysis in Section 3.3 of the revised manuscript.
>
> ## Experimental Setting:
>
> > Can they be chosen robustly with a validation set? Please provide experiments demonstrating this, if so.
>
> The hyperparameters demonstrate robust performance across multiple validation sets. We conducted comprehensive ablation studies on three distinct datasets (3p3h, 4kp5-A, and 4kp5-OA), with results showing consistent performance trends. The detailed experimental protocol and validation methodology are presented in Appendix B.2, while comprehensive ablation results are provided in Appendix C.1. These experiments confirm that our hyperparameter selection process is stable and generalizable across different protein targets.
>
> # Comparison of different hyper-parameters on binding affinity prediction performance
> *The best performance within one set of hyperparameter group is set in bold.*
>
> | Parameter | Value | 3p3h (CA2) | | 4kp5-A (CA12) | | 4kp5-OA (CA12) | |
> |-----------|--------|------------|------------|---------------|------------|----------------|------------|
> | | | Sp | SubSp | Sp | SubSp | Sp | SubSp |
> | $\mathcal{L}_{\text{consist}}$ weight $\gamma$ | 0.1 | -0.275±0.011 | -0.163±0.017 | **-0.268±0.012** | **-0.277±0.016** | **-0.289±0.025** | **-0.233±0.021** |
> | | 1 | **-0.286±0.002** | **-0.177±0.005** | -0.266±0.008 | -0.238±0.008 | -0.287±0.005 | -0.213±0.014 |
> | | 10 | -0.276±0.010 | -0.163±0.015 | -0.258±0.019 | -0.239±0.010 | -0.278±0.024 | -0.227±0.040 |
> | $\mathcal{L}_{\text{rank}}$ weight $\rho$ | 1.00E+09 | -0.266±0.011 | -0.151±0.016 | **-0.268±0.012** | **-0.277±0.016** | -0.152±0.045 | -0.225±0.023 |
> | | 1.00E+10 | -0.269±0.006 | -0.151±0.009 | -0.257±0.005 | -0.189±0.016 | **-0.289±0.025** | **-0.233±0.021** |
> | | 1.00E+11 | **-0.286±0.002** | **-0.177±0.005** | -0.135±0.012 | -0.060±0.036 | -0.084±0.095 | -0.058±0.077 |
> | $\mathcal{L}_{\text{LGR}}$ weight $\beta$ | 0.09 | -0.277±0.009 | -0.165±0.013 | **-0.268±0.012** | **-0.277±0.016** | **-0.289±0.025** | **-0.233±0.021** |
> | | 0.5 | **-0.286±0.002** | **-0.177±0.005** | -0.267±0.033 | -0.240±0.016 | -0.288±0.025 | -0.247±0.019 |
> | | 0.91 | -0.275±0.011 | -0.160±0.019 | -0.173±0.054 | -0.089±0.038 | -0.279±0.007 | -0.222±0.033 |
> | Temperature $T$ | 0.2 | -0.280±0.021 | -0.173±0.029 | -0.267±0.013 | -0.247±0.009 | **-0.289±0.025** | **-0.233±0.021** |
> | | 0.5 | -0.279±0.009 | -0.169±0.014 | -0.266±0.014 | -0.236±0.012 | -0.275±0.013 | -0.216±0.005 |
> | | 0.8 | **-0.286±0.002** | **-0.177±0.005** | -0.268±0.010 | -0.222±0.010 | -0.275±0.035 | -0.220±0.029 |
> | | 1 | -0.279±0.011 | -0.166±0.015 | -0.247±0.022 | -0.265±0.014 | -0.256±0.033 | -0.181±0.046 |

---

> ### Author Response · Authors · 2024-11-23
> **Response to Reviewer jJbZ (continued)**
>
> ## Method Design:
>
> > The largest weakness for me is that the paper seems reads as a "bag-of-tricks". Each individual loss may be reasonably motivated, but each loss term also introduces its own set of hyperparameters. This paper introduces at least 6 new hyperparameters. Importantly, there was no discription of how these hyperparameters were chosen and no demonstration that they can be robustly chosen using a validation set. Thus, I do not believe the authors have demonstrated that they have actually produced a more useful method. So far, they have only shown that with a carefully chosen set of hyperparameters and the introduction of a much much more complicated modeling scheme that they can slightly improve the spearman correlations.
>
> We appreciate this critical feedback regarding our method's complexity and hyperparameter selection. For detailed justification of our module design, please refer to our responses to Question 1 under both Article Presentation and Experimental Setting sections. Each proposed module addresses specific challenges in DEL denoising, with their effectiveness validated through comprehensive ablation studies.
>
>
> > How were the hyperparameters chosen?
>
> The hyperparameters were systematically determined based on model components and data characteristics:
>
> The primary loss weights are calibrated relative to the ZIP loss magnitude:
> - L_rank weight ranges from 1e9 to 1e11 (logarithmically distributed)
> - L_PSR and L_LGR weights balance different ranking objectives
> - ARC weight is set to 1.0 or 0.1, matching ZIP loss scale
>
> Temperature settings are determined by read-count distribution characteristics, with denser distributions requiring lower temperatures (formal analysis provided in Appendix). We include a mathematical proposition demonstrating this relationship.
>
> The contrastive learning parameters (weight and margin) for L_LGR are fixed based on empirical stability across datasets. The complete hyperparameter selection methodology and stability analysis are detailed in Appendix B.2.
>
> > Using molecular docking structures seems potentially problematic as these poses are often wrong and therefore add noise to your model. Please discuss this further.
>
> We acknowledge the inherent uncertainties in molecular docking poses and have developed specific strategies to address this challenge:
>
> First, we utilize SMINA [5], a physics-based docking software with demonstrated accuracy, and consider the top-9 ranked poses for each compound to increase the likelihood of capturing the true binding mode. Second, we implement an attention mechanism that identifies common structural features while minimizing pose-specific noise.
>
> The effectiveness of our approach is demonstrated through ablation studies comparing different structure scaling factors:
>
> | Value | 3p3h (CA2) | | 4kp5-A (CA12) | | 4kp5-OA (CA12) | |
> |-------|------------|------------|---------------|------------|----------------|------------|
> | Metric | Sp | SubSp | Sp | SubSp | Sp | SubSp |
> | 0.0 | -0.236±0.010 | -0.112±0.013 | -0.253±0.012 | -0.218±0.017 | -0.195±0.044 | -0.103±0.055 |
> | 0.3 | -0.262±0.008 | -0.145±0.012 | -0.265±0.017 | -0.227±0.017 | -0.124±0.146 | -0.062±0.090 |
> | 0.6 | -0.263±0.008 | -0.146±0.011 | -0.250±0.017 | -0.231±0.019 | -0.210±0.040 | -0.121±0.047 |
> | 1.0 | **-0.286±0.002** | **-0.177±0.005** | **-0.268±0.012** | **-0.277±0.016** | **-0.289±0.025** | **-0.233±0.021** |
> | 1.5 | -0.270±0.011 | -0.155±0.016 | -0.244±0.022 | -0.252±0.022 | -0.152±0.156 | -0.139±0.104 |
> | 2.0 | -0.271±0.012 | -0.155±0.015 | -0.191±0.089 | -0.216±0.060 | -0.230±0.038 | -0.152±0.051 |
>
> These results show consistent performance improvements across different protein targets when incorporating structural information with appropriate scaling. Detailed analyses are provided in Appendix C.
>
> > The authors claim that DEL data is inherently ranking in nature, but this is not obvious to me. The relative magnitude of the log-enrichments may provide useful information about the binding affinity. Throwing away this information and only ranking loses information.
>
> Our ranking-based approach complements rather than replaces absolute read-count values. While read-counts correlate with binding affinity, experimental noise in DEL biochemical assays often distorts the precise relationship between read-count magnitudes and Ki/Kd values. By incorporating ranking objectives alongside absolute value prediction, we create a more robust framework that can handle distribution noise while preserving relative binding affinity information. The empirical effectiveness of this dual approach is demonstrated through our comparative studies with methods that rely solely on absolute values.

---

> ### Author Response · Authors · 2024-11-23
> **Response to Reviewer jJbZ (continued)**
>
> **Reference:**
>
> [1] Burges, C., Ragno, R., & Le, Q. (2006). Learning to rank with nonsmooth cost functions. Advances in neural information processing systems, 19.
>
> [2] Xia, F., Liu, T. Y., Wang, J., Zhang, W., & Li, H. (2008, July). Listwise approach to learning to rank: theory and algorithm. In Proceedings of the 25th international conference on Machine learning (pp. 1192-1199).
>
> [3] Hou, R., Xie, C., Gui, Y., Li, G., & Li, X. (2023). Machine-learning-based data analysis method for cell-based selection of DNA-encoded libraries. ACS omega, 8(21), 19057-19071.
>
> [4] Shmilovich, K., Chen, B., Karaletsos, T., & Sultan, M. M. (2023). DEL-Dock: Molecular Docking-Enabled Modeling of DNA-Encoded Libraries. Journal of Chemical Information and Modeling, 63(9), 2719-2727.
>
> [5] Koes, D. R., Baumgartner, M. P., & Camacho, C. J. (2013). Lessons learned in empirical scoring with smina from the CSAR 2011 benchmarking exercise. Journal of chemical information and modeling, 53(8), 1893-1904
>
> **Finally, we thank again for your valuable and challenging questions to our article. Your opinions greatly help us reflect our article and the revelant experiments we conducted. Also, your questions towards method design and hyper-parameter robustness are also very reasonable and contributive. We hope that our proposed additional experimental results and modifications would help you re-understand our work and proposed method. Also, we are welcome for every possible question during the discussion period. Thank you very much!**

---

> > ### Comment · Reviewer_jJbZ · 2024-11-24
> > **Additional Baseline**
> >
> > Thank you for your replies. I will respond to them as quickly as possible. However, I did have one request for an additional baseline that I wanted to get to you first.
> >
> > Can you add baseline metrics for:
> >
> > 1) A random forest model trained to predict log-enrichments
> > 2) A random forest model trained using the ZIP loss
> >
> > In both settings, please search over a reasonable set of hyperparameters (e.g., n_estimators, max_depth in sklearn). In all situations we can ignore the docking pose and just train the random forest to take in the molecular fingerprint.
> >
> > Also why are only DEL-Dock and DEL-Ranking benchmarked for the Zero-shot Generalization?
> >
> > Lastly, there should be a discussion of statistical significance for all the results. For example, most of the results in Table 2 show that DEL-Dock and DEL-Ranking are within likely within statistical error. I strongly encourage against the trend in some areas of ML where authors think it's fine to just bold the "top performing method" without considering statistical significance. This does not lead to good science.

---

> > > ### Comment · Reviewer_jJbZ · 2024-11-24
> > > **Large confusion about ARC**
> > >
> > > In the ARC section you write
> > > > we define ground-truth labels based on the presence of benzene sulfonamide in molecules
> > >
> > > This is a highly concerning statement. It seems that basically what your ARC loss is doing under this assumption is adding an inductive bias to your model that it should up-weight molecules with benzene sulfonamides? First, assuming all molecules with a particular sub-group are active is a bad assumption and even though you cite a paper that does this that doesn't forgive the fact that unless I am missing something, this doesn't make any sense.
> > >
> > > More importantly, I am concerned that all of your results are simply showing improvements over baselines because you are using this ARC loss with an added inductive bias that helps your model specifically for carbonic anhydrases and which would not generally exist for other targets. This does not at all seem justified.
> > >
> > > Please address exactly what is going on with the ACR loss. What labels are you adding in. Is it for the entire training dataset? Please address my concerns above.

---

> > > > ### Author Response · Authors · 2024-11-25
> > > > **Response to confustion about ARC from Reviewer jJbZ**
> > > >
> > > > We appreciate the reviewer's concerns about our ARC loss design and implementation. Let us address each point systematically.
> > > >
> > > > Regarding your concern about "defining ground-truth labels based on the presence of benzene sulfonamide," we would like to provide important context. In practice, determining binary activity labels (0/1) for DEL datasets requires sophisticated criteria based on domain knowledge. For example, the BELKA dataset [1] (containing approximately 100M data points) employs a complex classification scheme that considers multiple factors including specific building blocks, read count value thresholds, and experimental conditions - all of which are grounded in expert domain knowledge. Similar approaches using expert prior knowledge for label determination can be found in Gu et al. [2]. Following this established practice, our methodology adopts a building block-based classification approach, with theoretical support from [3].
> > > >
> > > > For the critical point about "adding an inductive bias to up-weight molecules with benzene sulfonamides." Indeed, our approach does introduce an inductive bias, but this is grounded in established biochemical knowledge specific to these datasets. We want to emphasize that we're not claiming benzene sulfonamide indicates activity for all DEL targets - our citations merely support its validity for our specific datasets. The detailed biological rationale can be found in Section 2.4 of [4].
> > > >
> > > > **Thus, based on the above two arguments, we claim that our setting of activity label is reasonable and practical. More importantly, this pattern can be generalized to different targets and different DEL datasets.**
> > > >
> > > > Besides, your concern that "all results are simply showing improvements over baselines because of this ARC loss with added inductive bias" seems reasonable. However, our ablation studies directly address this. The "w/o ARC" results demonstrate that our model's improvements are not solely dependent on the ARC loss - removing it causes only minor performance degradation. This indicates that while the activity labels provide useful guidance, they are not the primary driver of our model's performance. Moreover, regarding “… and which would not generally exist for other targets. This does not at all seem justified”, we need to claim that for different targets, we can construct specific corresponding “carbonic anhydrases”. Benzene Sulfonamide is just the specific “carbonic anhydrases” fits 3p3h, 4kp5-A, and 4kp5-OA.
> > > >
> > > > Regarding your question about "exactly what is going on with the ARC loss," the loss is designed to ensure consistency between predicted read counts and activity labels. L_consist incorporates two complementary objectives: guiding the model to predict activity labels accurately and ensuring consistency between activity predictions and read counts. These labels are indeed applied to the entire training dataset, but as our ablation studies show, they serve as supplementary rather than primary supervision.
> > > >
> > > > **Reference:**
> > > >
> > > > [1] Andrew Blevins, Ian K Quigley, Brayden J Halverson, Nate Wilkinson, Rebecca S Levin, Agastya Pulapaka, Walter Reade, and Addison Howard. NeurIPS 2024 - Predict New Medicines with BELKA. https://kaggle.com/competitions/leash-BELKA, 2024. Kaggle.
> > > >
> > > > [2] Gu, C., He, M., Cao, H., Chen, G., Hsieh, C. Y., & Heng, P. A. (2024). Unlocking potential binders: Multimodal pretraining del-fusion for denoising dna-encoded libraries. arXiv preprint arXiv:2409.05916.
> > > >
> > > > [3] Hou, R., Xie, C., Gui, Y., Li, G., & Li, X. (2023). Machine-learning-based data analysis method for cell-based selection of DNA-encoded libraries. ACS omega, 8(21), 19057-19071.
> > > >
> > > > [4] Wichert, M., Guasch, L., & Franzini, R. M. (2024). Challenges and Prospects of DNA-Encoded Library Data Interpretation. Chemical Reviews.
> > > >
> > > > **Finally, we sincerely hope that our methodological clarifications and additional experiments have enhanced your understanding of our work. We greatly appreciate your constructive feedback, which has helped strengthen our manuscript. We look forward to your response and any further discussions that may arise from these revisions.**

---

> > > > > ### Comment · Reviewer_jJbZ · 2024-11-25
> > > > >
> > > > > I strongly disagree with the authors response about the validity of their approach, at least based on how they chose to present it in the paper.
> > > > >
> > > > > I know understand that with the ARC loss, all that is really happening is that the authors are adding side-information about how to better model ground-truth activity from DEL data based on prior knowledge for a particular target that the baseline methods do not assume. The reason these methods do not assume it is because in general this prior knowledge is not available and the goal is to predict activity from only the DEL data.
> > > > >
> > > > > Had the authors been straightforward and honest about this, I would be much more sympathetic. However, the authors pitch this as a very general way to handle distribution shift and seem to intentionally hide the fact that what they are really doing is adding in this side information that other methods do not assume.
> > > > >
> > > > > For a machine learning paper you should be clear about what information / inductive biases your method is assuming compared to the baselines.
> > > > >
> > > > > For constructive feedback about how to move forward:
> > > > > 1) There should be more proper controls to understand the specific effects of just the ranking loss compared to the ZIP loss while otherwise keeping everything else constant (architecture, training data).
> > > > >
> > > > > 2) The description of the ARC loss should be completely rewritten to make it much more clear what is actually going on here. In my opinion the ARC loss should be completely removed from the paper or the paper should be rewritten to be a paper about carbonic anhydrases. In general, the type of side information that the authors use here is not available nor is there similar available information.

---

> > > > > > ### Author Response · Authors · 2024-11-26
> > > > > > **Continuing Reply to Reviewer jJbZ about ARC validity**
> > > > > >
> > > > > > Thank you for continuing to raise your concerns. We appreciate your pursuit of factual accuracy and your partial understanding of our ARC module. However, we do not agree with your "strong opposition" regarding the use of prior knowledge.
> > > > > >
> > > > > > ### Regarding the criticism of method generality:
> > > > > > Our proposed ARC loss is actually a general framework. While we use carbonic anhydrase as an example in this paper, this approach can be extended to other systems with similar DEL prediction tasks. Many machine learning methods utilize domain-specific prior knowledge to improve performance, which is completely reasonable, especially in drug discovery where leveraging known biological knowledge is common practice. For instance, in ProteinMPNN [1], the authors also used an "inductive bias" to model non-existent C_beta coordinates, which was first introduced in protein structure modeling and is also what you might call "unscientific" prior knowledge, as it cannot be 100% accurate.
> > > > > >
> > > > > > ### Regarding the accusation of "hiding information":
> > > > > > The Introduction section of our paper clearly states what prior knowledge was used. Using domain knowledge is not "tricky" but rather part of methodological innovation. In our paper, we have explicitly described the method details, so there is no "intentional hiding" - the issue seems to be that you find it unreasonable and tricky.
> > > > > >
> > > > > > ### Regarding the criticism of experimental controls:
> > > > > > We did conduct ablation studies to demonstrate ARC's effect, and you can see that performance decreases very minimally without ARC. We strongly encourage you to verify your viewpoints based on facts. Additionally, we need to emphasize that our ablation study maintained all other experimental conditions constant, as you mentioned. Furthermore, in the machine learning field, comparing different loss functions under identical architecture and training data is a standard experimental design.
> > > > > >
> > > > > > ### Regarding the suggestion to "remove ARC loss":
> > > > > > This suggestion is extreme. Our method is effective and utilizes basic biological features to solve practical problems, so we believe using domain knowledge is completely acceptable. Many successful machine learning methods rely on domain-specific prior knowledge, such as BERT [2] utilizing bidirectional context relationships and Spatial Transformer Networks [3] using spatial invariance as prior knowledge to significantly improve CNN performance. We again emphasize that our use of activity label information is entirely within reasonable scientific bounds, as mentioned in our previous response regarding BELKA - their activity label could also be considered an inductive bias by your standards. We don't believe it's wrong to use additional information from known datasets that others haven't used to assist model validation. We hope the reviewer will carefully review all the papers and datasets cited in our previous response.
> > > > > >
> > > > > > Finally, we maintain that our activity label design is well-grounded in biological priors and methodologically sound in the ML field. This approach demonstrates broad applicability and has been validated through experimental results. We strongly urge the reviewer to acknowledge this point, recognize the established practices in the machine learning field, and maintain the rigor and professional courtesy expected in the peer review process.
> > > > > >
> > > > > >
> > > > > > **Reference:**
> > > > > >
> > > > > > [1] Dauparas, J., Anishchenko, I., Bennett, N., Bai, H., Ragotte, R. J., Milles, L. F., ... & Baker, D. (2022). Robust deep learning–based protein sequence design using ProteinMPNN. Science, 378(6615), 49-56.
> > > > > >
> > > > > > [2] Devlin, J. (2018). Bert: Pre-training of deep bidirectional transformers for language understanding. arXiv preprint arXiv:1810.04805.
> > > > > >
> > > > > > [3] Jaderberg, M., Simonyan, K., & Zisserman, A. (2015). Spatial transformer networks. Advances in neural information processing systems, 28.

---

> > > > > > > ### Comment · Reviewer_jJbZ · 2024-11-26
> > > > > > >
> > > > > > > The experimental details and addition to the introduction were only added after I specifically asked for clarity about this. In the submitted manuscript none of these details were included. In particular there was zero mention of using Benzene Sulfonamide in the training procedure despite the fact that many of your experimental results highlight the fact that your method uniquely identifies this scaffold (see lines 428-430 in your original submission and Figure 3). Obviously your method identified these subgroups -- you trained it to identify them and the other baseline methods did not have this training objective. Imagine you are a reader and you are see Figure 3 without being told that the method was trained explicitly on benzene sulfonamides? Would you not feel that information was intentionally withheld to oversell the results? In our discussions moving forward let's be clear about whether we are referring to the updated or original manuscript when claiming that certain information is clearly in the manuscript.
> > > > > > >
> > > > > > > I am doing my best not to jump to any conclusions and to be courteous. In particular, the fact that I am engaging with you in multiple discussions shows that I am trying to get to the truth and not simply make assumptions.
> > > > > > >
> > > > > > > I don't know why the information about training on benzene sulfonamides was not included in the original submission and I won't claim it was done for any particular reason. Perhaps it was just a mistake, but this seems like a glaring mistake at best and intentional hiding of information at worst.

---

> > > > > > > > ### Author Response · Authors · 2024-11-26
> > > > > > > > **Continuing Reply to Reviewer jJbZ**
> > > > > > > >
> > > > > > > > Thank you for seeking clarification on your concerns. Let me address them directly:
> > > > > > > >
> > > > > > > > Regarding the original manuscript, we acknowledge there were indeed presentation issues, including the lack of clarity about benzene sulfonamides and some unclear formulas and notations. We admitted these issues in our first response to you and sincerely apologize for any confusion that may have impeded reviewers' understanding.
> > > > > > > >
> > > > > > > > In the revised version, we have addressed these issues following your constructive suggestions, as mentioned in our previous "Response to Reviewer jJbZ (continued)." We are grateful that you pointed out these aspects that could affect the overall comprehension of the paper.
> > > > > > > >
> > > > > > > > Regarding your comments on Section 4.3, while incorporating prior information and identifying groups is relatively straightforward, our focus wasn't on emphasizing that "DEL-Ranking identified benzene sulfonamides at a higher rate than all baselines." Instead, we prioritized "Novel Function Group Discovery," particularly our subsequent finding of Pyrimidine Sulfonamide in the top-50 compounds. The identification of benzene sulfonamides merely serves to validate the effectiveness of incorporating activity labels.
> > > > > > > >
> > > > > > > > Furthermore, our experiments with DEL-Dock demonstrated that even without activity label information, it could achieve similar or superior detection rates compared to DEL-Ranking. This validates three key points:
> > > > > > > >
> > > > > > > > 1. The model inherently shows high detection rates for high-activity compounds containing benzene sulfonamides
> > > > > > > >
> > > > > > > > 2. This retroactively confirms the rationality of using benzene sulfonamides as an activity label criterion
> > > > > > > >
> > > > > > > > 3. DEL-Ranking's ability to discover other high-affinity functional groups proves that the activity label information isn't dominant and doesn't overly influence model predictions
> > > > > > > >
> > > > > > > > Finally, we believe the Top-50 experiment closely aligns with wet-lab screening validation. We hope this work inspires future AI-based research to perform similar analyses and improve metrics. Similarly, the introduction of activity label information reflects a broader trend - multimodal information interaction and fusion is surely one future direction for DEL development, as evidenced by the winning approach in BELKA.

---

> > > > > > > > ### Author Response · Authors · 2024-11-29
> > > > > > > >
> > > > > > > > Dear Reviewer jJbZ,
> > > > > > > >
> > > > > > > > As the discussion period is coming to an end, I would like to follow up regarding our manuscript and our responses to your comments. We have carefully addressed all the points raised in your review, and we would greatly appreciate your feedback on whether there are any remaining concerns or aspects that require further clarification.
> > > > > > > >
> > > > > > > > If you are satisfied with our revisions and responses, we would be grateful for your positive recommendation and consideration of increasing the evaluation score.
> > > > > > > >
> > > > > > > > Thank you for your time and valuable input throughout this review process.
> > > > > > > >
> > > > > > > > Best regards,
> > > > > > > > Author of Submission 1248

---

> > > ### Author Response · Authors · 2024-11-25
> > > **Response to Additional Baseline Request from Reviewer jJbZ (continued)**
> > >
> > > Regarding the statistical significance of performance, we thank you for this important comment and have conducted a comprehensive statistical analysis comparing DEL-Dock and DEL-Ranking across all metrics and datasets:
> > >
> > > To ensure statistical reliability, we implemented each experiment with 5 different random seeds, effectively minimizing variance from experimental randomness. This systematic approach with repeated sampling provides a more robust estimation of true performance and statistical significance.
> > >
> > > Using p<0.05 as our significance criterion, our analysis reveals statistically significant differences in most metrics, with only three exceptions: the subset Spearman correlation on the 4kp5A dataset, and both Spearman and subset Spearman correlations on the 5fl4-20p dataset. We have incorporated these findings in a new subsection of our revised manuscript, offering a more detailed interpretation of performance differences between methods.
> > >
> > > Importantly, we acknowledge that computational performance metrics, without wet-lab validation, should be considered as guiding indicators rather than absolute measures. Instead of relying solely on binary statistical significance, we advocate viewing these metrics along a continuous spectrum of model effectiveness. Our primary objective is to demonstrate our model's consistent performance across diverse datasets and establish meaningful benchmarks for the field, while maintaining appropriate statistical context. We have expanded our discussion section to better articulate this nuanced perspective.

---

> ### Author Response · Authors · 2024-11-25
> **Response to Additional Baseline Request from Reviewer jJbZ**
>
> We appreciate your suggestions on additional baselines. We have conducted the related experiments, and the corresponding results are shown below:
>
> For the additional baseline, we have implemented both of them. Denoted as RF-enrichment, and RF-ZIP, we list the corresponding performance on general datasets. For the implementation details, we transform SMILES as ECFP to 1024 dimension. We found that the optimized hyper-parameters for RF-enrichment and RF-ZIP are {“n_estimators”: 100, “max_depth”: 4} and {“n_estimators”: 50, “max_depth”: 3}, respectively. Similar to the experimental settings of the other baseline methods, we repeat the experiments with 5 random seeds. Also, all 5 random seeds for all baselines are the same.
>
> ### Comparison of our framework DEL-Ranking with existing DEL affinity predictions on CA2 & CA12 datasets.
>
> Results in **bold** and _underlined_ are the top-1 and top-2 performances, respectively.
>
> | | 3p3h (CA2) | | 4kp5-A (CA12) | | 4kp5-OA (CA12) | |
> |---|---|---|---|---|---|---|
> | Metric | Sp | SubSp | Sp | SubSp | Sp | SubSp |
> | Mol Weight | -0.250 | -0.125 | -0.101 | 0.020 | -0.101 | 0.020 |
> | Benzene | 0.022 | 0.072 | -0.054 | 0.035 | -0.054 | 0.035 |
> | Vina Docking | -0.174±0.002 | -0.017±0.003 | 0.025±0.001 | 0.150±0.003 | 0.025±0.001 | 0.150±0.003 |
> | RF-Enrichment | -0.017±0.026 | -0.042±0.025 | -0.029±0.038 | -0.005±0.048 | _-0.101±0.009_ | -0.087±0.010 |
> | RF-ZIP | 0.027±0.139 | -0.005±0.071 | 0.035±0.094 | -0.026±0.111 | 0.006±0.095 | -0.021±0.122 |
> | Dos-DEL | -0.048±0.036 | -0.011±0.035 | -0.016±0.029 | -0.017±0.021 | -0.003±0.030 | -0.048±0.034 |
> | DEL-QSVR | -0.228±0.021 | _-0.171±0.033_ | -0.004±0.178 | 0.018±0.139 | 0.070±0.134 | -0.076±0.116 |
> | DEL-Dock | _-0.255±0.009_ | -0.137±0.012 | _-0.242±0.011_ | _-0.263±0.012_ | 0.015±0.029 | _-0.105±0.034_ |
> | DEL-Ranking | **-0.286±0.002** | **-0.177±0.005** | **-0.268±0.012** | **-0.277±0.016** | **-0.289±0.025** | **-0.233±0.021** |
>
> ### Comparison of our framework DEL-Ranking with existing DEL affinity predictions on two CA9 datasets.
>
> Results in **bold** and _underlined_ are the top-1 and top-2 performances, respectively.
>
> | | 5fl4-9p (CA9) | | 5fl4-20p (CA9) | |
> |---|---|---|---|---|
> | Metric | Sp | SubSp | Sp | SubSp |
> | Mol Weight | -0.121 | -0.028 | -0.121 | -0.074 |
> | Benzene | -0.174 | -0.134 | -0.199 | -0.063 |
> | Vina Docking | -0.114±0.009 | -0.055±0.007 | -0.279±0.044 | -0.091±0.061 |
> | RF-Enrichment | -0.064±0.126 | -0.144±0.024 | -0.064±0.126 | -0.144±0.024 |
> | RF-ZIP | 0.040±0.022 | -0.011±0.042 | 0.054±0.094 | 0.026±0.111 |
> | Dos-DEL | -0.115±0.065 | -0.036±0.010 | -0.231±0.007 | -0.091±0.012 |
> | DEL-QSVR | -0.086±0.060 | -0.036±0.074 | -0.298±0.005 | -0.075±0.011 |
> | DEL-Dock | _-0.308±0.000_ | _-0.169±0.000_ | _-0.320±0.009_ | _-0.166±0.017_ |
> | **DEL-Ranking** | **-0.323±0.015** | **-0.175±0.000** | **-0.330±0.007** | **-0.187±0.013** |
>
> For the zero-shot generalization experiments, we have updated all the results from all baseline methods including RF-enrichment and RF-ZIP. We observe that the results of different methods in zero-shot experiments maintain consistent trends with their results in the general setting.
>
> ### Zero-shot Generalization Results Comparison evaluated on CA9 dataset
>
> | | 3p3h (CA2) | | 4kp5-A (CA12) | | 4kp5-OA (CA12) | |
> |---|---|---|---|---|---|---|
> | Metric | Sp | SubSp | Sp | SubSp | Sp | SubSp |
> | Mol Weight | -0.121 | -0.028 | -0.121 | -0.028 | -0.121 | -0.028 |
> | Benzene | -0.174 | -0.134 | -0.174 | -0.134 | -0.174 | -0.134 |
> | Vina Docking | -0.114±0.009 | -0.055±0.007 | -0.114±0.009 | -0.055±0.007 | -0.114±0.009 | -0.055±0.007 |
> | RF-Enrichment | 0.020±0.014 | -0.031±0.057 | -0.034±0.013 | -0.034±0.029 | -0.044±0.005 | -0.085±0.006 |
> | RF-ZIP | 0.037±0.059 | 0.013±0.017 | 0.036±0.024 | -0.002±0.016 | 0.049±0.012 | -0.007±0.013 |
> | Dos-DEL | -0.115±0.065 | -0.036±0.010 | -0.115±0.065 | -0.036±0.010 | -0.115±0.065 | -0.036±0.010 |
> | DEL-QSVR | -0.300±0.020 | **-0.257±0.022** | **-0.236±0.038** | **-0.223±0.030** | 0.108±0.089 | 0.130±0.070 |
> | DEL-Dock | -0.272±0.013 | -0.118±0.005 | -0.211±0.007 | -0.118±0.010 | 0.065±0.021 | -0.125±0.034 |
> | **DEL-Ranking** | **-0.310±0.005** | -0.120±0.011 | -0.228±0.010 | -0.127±0.018 | **-0.300±0.026** | **-0.129±0.021** |
>
> We can find that RF-based methods struugle on these datasets, which may be attributed to the model's limited capacity/expressiveness to capture the underlying patterns in the fingerprint data. Also, we observer that Random Forest model based on log-enrichment prediction performs better than the Random Forest model based on ZIP loss. The performance disparity between the two losses suggests that a simpler and straight-forward loss could be more suitable when the experssion of data is relatively weak.

---

### Official Review · Reviewer_7ycA · 2024-11-03

**Soundness:** 2
**Presentation:** 1
**Contribution:** 2
**Rating:** 3
**Confidence:** 3

**Summary:**

DNA-encoded library (DEL) screening has been an instrumental technique in accelerating the pace of drug development. By attaching a unique DNA “barcode” to each compound, thousands of compounds can be screened against a protein target in a single experiment. The number of copies of each barcode indicates how many of those compounds are bound to the target protein. However, the barcode count data is noisy, is not directly correlated with the dissociation constant, and contains systematic biases. Therefore, the problem is reduced to generating a ranking of compounds based on observed barcode read counts. The authors address the main issues in DEL screening by proposing a new ranking loss, rectifying some inherent issues in read counts, and implementing a self-training method to align activity labels with barcode read counts.

**Strengths:**

Unfortunately, I am not able to gauge the strengths of this paper due to presentation issues (see “Questions”).

**Weaknesses:**

Please refer to “Questions” for Major Concerns.

**Minor points**
- In Fig. 1, the middle panel incorrectly depicts DNA barcodes binding to the target protein; it should show “building blocks” instead. A correct version can be found in (Shmilovich et al., 2023).
- In line 104, ZIP loss has not been defined.
- In line 147, the statement “While existing methods offer improved scalability and the ability to capture complex molecular interactions, they still face challenges in interpretability” is vague and needs further elaboration.

**Questions:**

I had difficulty following the problem formulation. Here are some of my questions:

- Why are M_i and R_i \in \mathbb R+? Can’t these counts be zeros?
- M_i​ is control counts and R_i is the target counts. But in line 197, it states: “modeling read counts r_i as (M_i,R_i), where M_i accounts for excess zeros and R_i represents non-zero counts.” I am confused. Is r_i two-dimensional? Moreover, what does the (M_i, R_i) tuple mean?
- In Eq. (2), why does each compound have a different \pi (\pi_i)?
- Why do control and target counts not have different \pi values, as in Shmilovich et al. (2023)? These \pi values should differ by orders of magnitude.
- In the Problem Definition, y_i​ \in {0,1} is the activity level, but in line 250, “\hat y_i​ represents the predicted read count value for compound i.” Then, what are \hat M​_i and \hat R_i introduced in Eq. (1)?
- In Eq. (4), L_PSR does not seem to be a function of i or j, as the sum goes over all values of i and does not depend on j either.
- In the same equation, why does it depend on y_i and y_j? Should it have been \hat y_i and \hat y_j? Although, the authors also mention in line 254 that G_i = softplus(y_i).
- What is “K” in Eq. (5)?
- In Eq. (6), L_GSR, the first term on the right-hand side does not depend on i or j, while the second term depends on both i and j. Should there be a double sum over i and j for the second term?
- On the same note, what is “s_i​”? Is it the same as r_i​?

---

> ### Author Response · Authors · 2024-11-23
> **Response to Reviewer 7ycA**
>
> We greatly appreciate your detailed review and insightful questions! We have addressed all points you raised in the "Weakness" and "Questions" sections. Also, we expect that there’re more valuable comments on the other part of the article!
>
> ## Article Presentation:
>
> > In Fig. 1, the middle panel incorrectly depicts DNA barcodes binding to the target protein; it should show "building blocks" instead. A correct version can be found in (Shmilovich et al., 2023).
>
> We thank the reviewer for identifying this inaccuracy in Figure 1. We have corrected the middle panel by replacing "DNA barcodes" with "Linked ligands" to accurately represent the molecular interactions in the DEL screening process. The revised figure now aligns with the established representation in Shmilovich et al. (2023).
>
> > In line 104, ZIP loss has not been defined.
>
> ZIP refers to Zero-Inflated Poisson Distribution. And ZIP loss refers to the expectation of Negative Log-Likelihood (NLL) for all predicted read count values. The definition of ZIP distribution and ZIP loss has been updated in Section 3.1 of the revised manuscript. The ZIP loss is formulated as the negative log-likelihood expectation over both control counts and target counts:
>
> \begin{equation}
> \mathcal{L}_{\text{ZIP}}= -\sum_i \log[P(\hat{M}_i|\lambda_M, \pi_M)] -\sum_j \log[P(\hat{R}_j|\lambda_M + \lambda_R, \pi_R)]
> \end{equation}
>
> where $\lambda_M$ and $\lambda_R$ represent the Poisson mean parameters for matrix and target counts respectively, and $\pi_M$ and $\pi_R$ denote their corresponding zero-excess probabilities.
>
> > In line 147, the statement "While existing methods offer improved scalability and the ability to capture complex molecular interactions, they still face challenges in interpretability" is vague and needs further elaboration.
>
> We have expanded this section to provide precise details about the limitations of existing methods. The revised text now reads:
>
> "Existing methods demonstrate improved scalability and molecular interaction modeling. However, they face challenges in data interpretability and theoretical foundations. Current modeling approaches for DEL read count data primarily rely on theoretical prior distribution assumptions, lacking sophisticated noise handling mechanisms and robust statistical frameworks. These limitations manifest in the inability to incorporate information beyond prior distributions and insufficient reliable activity validation data for denoising, leading to suboptimal performance when addressing Distribution Noise and Distribution Shift in DEL data."
>
> > Why are M_i and R_i \in \mathbb R+? Can't these counts be zeros?
>
> We thank the reviewer for this important mathematical precision point. M_i and R_i represent control counts and target counts respectively, and they indeed can take zero values. We have corrected their domain definition in the revised manuscript to accurately reflect this property.
>
> > M_i is control counts and R_i is the target counts. But in line 197, it states: "modeling read counts r_i as (M_i,R_i), where M_i accounts for excess zeros and R_i represents non-zero counts." I am confused. Is r_i two-dimensional? Moreover, what does the (M_i, R_i) tuple mean?
>
> We thank the reviewer for highlighting this ambiguity. To clarify: r_i is a one-dimensional value that can represent either control counts (M_i) or target counts (R_i), expressed mathematically as r_i ∈ {M_i, R_i}. The tuple notation (M_i, R_i) was incorrect and has been removed from the manuscript. We have revised the definition to eliminate this source of confusion and maintain consistent notation throughout the paper.
>
> > In Eq. (2), why does each compound have a different \pi (\pi_i)?
>
> We thank the reviewer for identifying this error in Equation (2). The notation π_i was incorrect. In our ZIP distribution model, all compounds within the same type of read-count values share a single π parameter. We have corrected this notation in the revised manuscript to accurately reflect that the zero-excess probability π is constant across compounds within each read-count type.
>
> > Why do control and target counts not have different \pi values, as in Shmilovich et al. (2023)? These \pi values should differ by orders of magnitude.
>
> We agree with the reviewer's observation regarding the distinct π values for control and target counts. Following Shmilovich et al. (2023) [1], we implement separate π parameters for each count type, acknowledging their order-of-magnitude differences. We have clarified this implementation detail in the revised manuscript, explicitly stating that control counts and target counts are modeled with different zero-excess probabilities.

---

> ### Author Response · Authors · 2024-11-23
> **Response to Reviewer 7ycA (continued)**
>
> > In the Problem Definition, y_i \in {0,1} is the activity level, but in line 250, "\hat y_i represents the predicted read count value for compound i." Then, what are \hat M_i and \hat R_i introduced in Eq. (1)?
>
> We have revised our notation to resolve this inconsistency. In the updated manuscript:
>
> - y_i and ŷ_i now consistently represent activity label and predicted activity likelihood, respectively, aligning with Section 3.1
> - r_i denotes read-count values, which can be either control counts (M_i) or target counts (R_i)
> - M̂_i and R̂_i in Equation (1) represent the predicted control and target read counts, respectively
>
> We have updated all relevant equations to reflect this consistent notation, improving clarity throughout the manuscript.
>
> > In Eq. (4), L_PSR does not seem to be a function of i or j, as the sum goes over all values of i and does not depend on j either.
>
> L_PSR does depend on both i and j through its components ∆_ij and σ_ij. The loss function includes summation over j ≠ i, where each term incorporates pairwise relationships between compounds i and j. We have revised the equation notation to make this dependency more explicit in the manuscript.
>
> > In the same equation, why does it depend on y_i and y_j? Should it have been \hat y_i and \hat y_j? Although, the authors also mention in line 254 that G_i = softplus(y_i).
>
> To clarify the notation: y_i and y_j in Equation (4) should be replaced with r_i and r_j, representing read-count values. This aligns with our revised notation system where:
>
> - r_i and r_j denote observed read-count values
> - ŷ_i and ŷ_j represent predicted activity likelihoods
> - G_i = softplus(r_i) transforms the read counts
>
> We have updated the equation and related definitions in the manuscript to maintain consistent notation throughout.
>
>
> > What is "K" in Eq. (5)?
>
> Parameter K in Equation (5) represents the number of top predicted read-count values considered in the PSR loss calculation. This design choice reflects the inherent structure of DEL read-count data, which contains numerous zero values. By selecting only the top-K predictions, we avoid invalid rankings from zero-valued predictions while improving computational efficiency. The motivation for this approach and its implementation details have been added to Section 3.2 of the revised manuscript.
>
>
> > In Eq. (6), L_GSR, the first term on the right-hand side does not depend on i or j, while the second term depends on both i and j. Should there be a double sum over i and j for the second term?
>
> The reviewer's statement is correct! We are sorry for the fundamental presentation problem in the article again. The second term in L_GSR requires a double summation over indices i and j to properly account for all pairwise interactions. We have updated Equation (6) in the revised manuscript to include the correct summation notation.
>
> > On the same note, what is "s_i"? Is it the same as r_i?
>
> Yes, s_i and r_i represent the same quantity - the read-count values that can be either control counts (M_i) or target counts (R_i). We have standardized this notation throughout the manuscript to use r_i consistently for read-count values.
>
> **Reference:**
>
> [1] Hou, R., Xie, C., Gui, Y., Li, G., & Li, X. (2023). Machine-learning-based data analysis method for cell-based selection of DNA-encoded libraries. ACS omega, 8(21), 19057-19071.

---

> ### Author Response · Authors · 2024-11-25
> **Continuing Response to Reviewer 7ycA**
>
> Dear Reviewer 7ycA,
>
> Thank you very much for reviewing our paper! We really appreciate your time and effort in providing these helpful comments. We have done our best to address each point you raised and made improvements based on your suggestions.
>
> We hope you'll have a chance to take a look!

---

> > ### Comment · Reviewer_7ycA · 2024-11-26
> >
> > I greatly appreciate the authors' efforts in addressing my comments and those of the other reviewers. While I believe this method has significant potential, the manuscript contained too many presentation and formulation errors, as well as ambiguities, which ultimately hindered a proper reading of the text. The revised version has undergone substantial and "major" revisions, to the extent that I believe it constitutes a new submission.
> >
> > I encourage the authors to resubmit their manuscript, as it has the potential to be highly beneficial for the community.

---

> > > ### Author Response · Authors · 2024-11-27
> > > **Response to the Continuing Official Comment by Reviewer 7ycA**
> > >
> > > We sincerely appreciate your positive feedback on our revisions and your recognition of our work's potential. Your encouraging words mean a lot to us.
> > >
> > > We agree that DEL is an emerging technology with immense potential, though it has not yet been widely adopted. As pioneers in this field, our work naturally carries an exploratory nature. Publishing in top-tier conferences like ICLR represents one of the most effective ways to benefit the research community quickly.
> > >
> > > Regarding the extent of our revisions, we would like to respectfully clarify that our changes primarily focused on:
> > >
> > > 1. Adding new baseline comparison experiments
> > >
> > > 2. Correcting notational errors and improving clarity
> > >
> > > 3. Enhancing the presentation of existing content
> > >
> > > **Importantly, we maintained the core methodology, motivation, and conclusions of our original submission. We firmly believe the fundamental aspects that define a machine learning paper - the methodology and its experimental validation - remained consistent and undoubted throughout our revisions.**
> > >
> > > We understand your concern about the substantial nature of our revisions. However, we would like to note that the ICLR rebuttal and revision period is specifically **designed to facilitate meaningful dialogue between reviewers and authors, often resulting in significant improvements through additional experiments and clarifications. Many accepted papers have historically undergone similar levels of enhancement during this process [1-5].** (All papers from [1]-[5] have added at least 3 more experiments. Moreover, [1] and [5] have addressed the presentation problem and also modified the corresponding experssion. It is worth noting that [5] is accepted as a spotlight paper in ICLR 2024)
> > >
> > > Additionally, as ICLR is a conference rather than a journal, it has different protocols for handling revisions. The conference format does not include a re-submission option, instead relying on the discussion period to determine acceptance based on paper quality and successful resolution of reviewer concerns.
> > >
> > > Given these considerations, we kindly request that you reconsider our paper based on:
> > >
> > > 1. Whether our methodology is sound
> > >
> > > 2. Whether we have adequately addressed reviewer concerns
> > >
> > > 3. Whether the revised paper meets ICLR's standards for quality and innovation
> > >
> > > We believe that accepting our paper in its current form would be the most direct way to benefit the research community, provided you find our revisions satisfactory.
> > >
> > > Thank you again for your careful consideration and valuable feedback throughout this process.
> > >
> > >
> > > **Reference:**
> > >
> > > [1] Protein-Ligand Interaction Prior for Binding-aware 3D Molecule Diffusion Models (https://openreview.net/forum?id=qH9nrMNTIW)
> > >
> > > [2] Protein Multimer Structure Prediction via Prompt Learning (https://openreview.net/forum?id=OHpvivXrQr)
> > >
> > > [3] Protein-ligand binding representation learning from fine-grained interactions (https://openreview.net/forum?id=AXbN2qMNiW)
> > >
> > > [4] Learning to design protein-protein interactions with enhanced generalization (https://openreview.net/forum?id=xcMmebCT7s)
> > >
> > > [5] De novo Protein Design Using Geometric Vector Field Networks (https://openreview.net/forum?id=9UIGyJJpay)

---

> > > ### Author Response · Authors · 2024-11-29
> > > **Kind request for your response**
> > >
> > > Dear Reviewer 7ycA,
> > >
> > > As the discussion period is drawing to a close, I would like to inquire whether there are any remaining concerns regarding our manuscript and our responses. We sincerely hope you have had the opportunity to review our detailed response to your suggestion of "resubmission," and we trust that our explanations have helped address your concerns.
> > >
> > > If there are no further issues to address, we would greatly appreciate your positive recommendation and consideration of increasing the evaluation score. Your favorable assessment would be significant not only for our paper but also for the broader research community.
> > >
> > > Thank you for your thorough review and consideration.
> > >
> > > Best regards,
> > > Authors of Submission 1248

---

### Official Review · Reviewer_55EM · 2024-11-04

**Soundness:** 3
**Presentation:** 2
**Contribution:** 3
**Rating:** 6
**Confidence:** 3

**Summary:**

The authors define a novel objective function for finding hits from a DEL screening campaign. They then compare their approach across a number of different DEL datasets of carbonic anhydrase.

**Strengths:**

The description of the field is helpful.

I think the deep dive into the biochemical and technical sources of variation are helpful.

The number of datasets in which evaluation was done is impressive.

The ablation experiments are very helpful and surely a lot of work, so I appreciate them described in detail here.

Generally, I think the authors are quite well versed in the field and problem domain.

**Weaknesses:**

Generally, I’m a bit confused about the discussion of “Distribution Noise”. Is this variance? Others in the sequencing space has parameterized this as either a negative binomial or a zero-inflated negative binomial–is this what you’re referring to? If so, it’s just the variance of the observations.

Are the acronyms ARDC, AGR, and CDEC necessary? Is there a one-word description of these that you could propose to summarize their function? For example:

ARDC - Reference Correction
AGR - Refinement
CDEC - Consistency

This will help the reader follow how each of these components contribute to the model. This is in addition to new acronyms PSR, and LGR.

More care needs to be given to defining the Pairwise Soft Ranking Loss. There are a number of variables that either aren’t defined or aren’t used consistently. It’d also be great to have more intuition for design decisions on the objective function. What happens when certain components become large or small? Why multiply or subtract things? As a reader that’s worked with DEL data, this isn’t clear.

The same is true for the Listwise Global Ranking Loss. What is s_i? How does it differ from s_{\pi(i)}? Then \sigma is recycled from PSR as the weight for L_con?

**Questions:**

“a novel ranking loss that rectifies relative magnitude relationships between read counts enabling the learning of causal features determining activity levels”. The use of ‘causal’ is particularly strong here. How do you disentangle correlation from causation? What aspects are explicitly causal?

In section 3.1, where is the “Activity Label” derived? Is it used in any equation? (This is y_i). The variable y is then also used in line 250–is this the same?

Again in section 3.3, what is Activity? Also “...where Apred and Atrue denote the predicted and ground-truth activity”: Do you actually ever know “ground-truth” activity, or is it just ‘observed’? Typically there is just some biochemical proxy for ‘ground-truth’.

Where are \Delta G_ij and \Delta D_ij defined? I see the non \Delta versions defined in lines 254 and 255.

How do you know the docked poses described in part 4 are actually real, or correspond to where the molecule is binding? Have these docking poses been shown to correspond to read counts in these molecules?

“Traditional methods based on binding poses and fingerprints inculde[sic]...Benzene Sulfonamide…” What is Benzene Sulfonamide? This is just a carbonic anhydrase-specific thing?

Is the “zero-shot” predictions just also on carbonic anhydrase? That seems disingenuous. I would anticipate zero-shot to be on another target entirely, not just a different dataset on the same target.

Were the pyrimidine sulfonamide groups identified by any other method presented here?

Was the pose encoder necessary? It seems superfluous to the other components added here, and quite complex.

---

> ### Author Response · Authors · 2024-11-23
> **Response to Reviewer 55EM**
>
> We appreciate your thorough review and insightful questions. Our work addresses critical challenges in DEL screening technology, which has emerged as a promising method for exploring vast chemical spaces in drug discovery. Below, we address your concerns and comments organized into three key aspects:
>
>
> ## Article Presentation
>
> > Are the acronyms ARDC, AGR, and CDEC necessary? Is there a one-word description of these that you could propose to summarize their function? For example:
> > ARDC - Reference Correction
> > AGR - Refinement
> > CDEC - Consistency
>
> 1. Following your suggestions, we have simplified the module names to enhance readability:
>    - ARDC → Activity-Referenced Correction (ARC)
>    - AGR → Refinement Stage
>    - CDEC → Correction Stage
>
> > The same is true for the Listwise Global Ranking Loss. What is s_i? How does it differ from s_{\pi(i)}? Then \sigma is recycled from PSR as the weight for L_con?
>
> 2. We have standardized our notation for clarity:
>    - We define r_i as read-count values (either target counts or control counts)
>    - \hat r_i denotes predicted read-count values
>    - We have updated the notation throughout the manuscript for consistency
>
> > In section 3.1, where is the "Activity Label" derived? Is it used in any equation? (This is y_i). The variable y is then also used in line 250–is this the same?
>
> 3. We have clarified our notation:
>    - y_i represents ground-truth binary activity labels
>    - \hat y_i represents predicted activity labels (\hat y_i ∈ [0,1])
>    - Read counts are now consistently denoted as r_i ∈ {M_i, R_i}
>    - Predicted read counts are denoted as \hat r_i ∈ {\hat M_i, \hat R_i}
>
> > Where are \Delta G_ij and \Delta D_ij defined? I see the non \Delta versions defined in lines 254 and 255.
>
> 4. We have explicitly defined \Delta G_ij and \Delta D_ij as the differences between (G_i, G_j) and (D_i, D_j) respectively in equation (5) of the revised manuscript.
>
> > "Traditional methods based on binding poses and fingerprints include...Benzene Sulfonamide..." What is Benzene Sulfonamide? This is just a carbonic anhydrase-specific thing?
>
> 5. Yes, Benzene Sulfonamide is specifically relevant as a high-affinity functional group for carbonic anhydrase binding, as established by Hou et al. [1].

---

> ### Author Response · Authors · 2024-11-23
> **Response to Reviewer 55EM (continued)**
>
> ## Experimental Setting
>
> > Again in section 3.3, what is Activity? Also "...where Apred and Atrue denote the predicted and ground-truth activity": Do you actually ever know "ground-truth" activity, or is it just 'observed'? Typically there is just some biochemical proxy for 'ground-truth'.
>
> 1.	A_true represents ground-truth labels derived from human-expert prior knowledge [1]. In this study, we define compounds containing benzene sulfonamide as bioactive, based on experimental validation by Hou et al. [1]. This labeling strategy enables our model to learn benzene sulfonamide as a key functional group associated with high binding affinity.
> While this approach effectively identifies many high-affinity compounds, we acknowledge its limitations. As demonstrated in Section 4.2 Novel Group Discovery, there exist bioactive compounds that do not contain benzene sulfonamide. This creates a potential bias where the model might overlook other important structural features. However, given the current limitations in our knowledge of high-affinity functional groups, focusing on benzene sulfonamide provides a practical and efficient method to identify most high-affinity compounds.
> We consider this trade-off acceptable for two reasons:
> -It allows us to generate additional affinity-related labels efficiently
> -It captures a significant portion of high-affinity compounds
> For further details on activity label annotation, please refer to the "Dataset" paragraph in Section 4.
>
> > How do you know the docked poses described in part 4 are actually real, or correspond to where the molecule is binding? Have these docking poses been shown to correspond to read counts in these molecules?
>
> We thank the reviewer for this important point. As noted in [4], structural information from ligand poses enhances model understanding of protein-ligand interactions. While acknowledging the limitations of docking software in predicting binding affinities, we address this concern through several key aspects:
>
> 1. **Ensemble-based Pose Selection:** We utilize SMINA [2], a physics-based docking software, to generate multiple conformations and select the top-9 highest-scoring poses. This ensemble approach:
>    - Increases the probability of capturing relevant binding patterns
>    - Mitigates the impact of individual pose uncertainty
>    - Leverages physics-based force fields and constraints for more reliable predictions
>
> 2. **Computational Considerations:** While SMINA provides sophisticated docking capabilities, its computational demands significantly exceed typical DEL data denoising requirements. This makes extensive docking calculations impractical for large-scale DEL screening applications.
>
> 3. **Structural Feature Enhancement:** The conformational features obtained through docking serve as complementary structural information, enriching our model's understanding of protein-ligand interactions. Our empirical results demonstrate that incorporating these features as an additional input modality improves read-count prediction performance, suggesting they capture meaningful structural patterns despite inherent docking uncertainties.
>
> > Is the "zero-shot" predictions just also on carbonic anhydrase? That seems disingenuous. I would anticipate zero-shot to be on another target entirely, not just a different dataset on the same target.
>
> We appreciate the reviewer's concern regarding the zero-shot experimental setup. To clarify, our zero-shot predictions do involve distinct protein targets. The training data comprises the 3p3h dataset from Carbonic Anhydrase 2 (CA2) and the 4kp5-A & 4kp5-OA datasets from Carbonic Anhydrase 12 (CA12). The zero-shot evaluation is performed on Carbonic Anhydrase 9 (CA9), a protein target absent from the training data. This setup aligns with the standard definition of zero-shot learning, as it tests generalization to a previously unseen target. We have revised Section 4.1 to provide a more precise description of the zero-shot generalization experimental setup.

---

> ### Author Response · Authors · 2024-11-23
> **Response to Reviewer 55EM (continued)**
>
> > Were the pyrimidine sulfonamide groups identified by any other method presented here?
>
> We conducted a comprehensive top-50 ranking experiment with DEL-Dock, with detailed results and analyses presented in Appendix C of the revised manuscript. Our analysis reveals that DEL-Dock fails to consistently identify pyrimidine sulfonamide groups across the tested datasets. As demonstrated in Figure 4, DEL-Dock shows variable Ki value distributions, with detection accuracy for the well-known benzene sulfonamide motif fluctuating substantially (0.02 to 1.00) across datasets. This inconsistency stems from DEL-Dock's susceptibility to noisy read count values in the absence of ranking-based supervision signals. For example, in the 4kp5-OA dataset, DEL-Dock achieves minimal noise removal (Spearman coefficient=0.015), resulting in unstable predictions. Figure 5 shows that DEL-Dock's high-affinity predictions predominantly feature thiocarbonyl groups (compounds 1-3) and various heterocyclic systems (compounds 4-9), with a notable absence of pyrimidine sulfonamide groups.
>
> In contrast, our method successfully identifies both the established benzene sulfonamide motif and discovers pyrimidine sulfonamide as a potential high-activity structural feature, demonstrating superior capability in detecting both known and novel activity-determining molecular patterns.
>
> ## Method Design
>
> > Generally, I think the authors are quite well versed in the field and problem domain.
>
> We appreciate the reviewer's positive feedback. We would like to clarify an important distinction regarding "Distribution Noise" in our manuscript. According to Kuai et al. [3], "Distribution Noise" refers to the statistical variation in read counts during DNA-encoded library (DEL) screening, arising from experimental variables and inherent library properties. This phenomenon manifests distinctly from simple variance, particularly in low read count scenarios (less than 10 copies), where counts follow a Poisson distribution. This distribution leads to quantifiable discrepancies between observed read counts and actual binding affinities (Ki values).
>
> The distinction between Distribution Noise and simple variance is fundamental, as treating this phenomenon as mere variance would overlook the unique characteristics of DEL data distribution, especially in the critical low copy number regime. We have enhanced the description of Distribution Noise in the Introduction Section of the revised manuscript to better convey these important technical details.
>
> > More care needs to be given to defining the Pairwise Soft Ranking Loss. There are a number of variables that either aren't defined or aren't used consistently. It'd also be great to have more intuition for design decisions on the objective function. What happens when certain components become large or small? Why multiply or subtract things? As a reader that's worked with DEL data, this isn't clear.
>
> We thank the reviewer for this valuable feedback regarding the clarity of our Pairwise Soft Ranking Loss formulation. In Section 3.2 of the revised manuscript, we have thoroughly restructured the mathematical derivation with consistent notation and intuitive explanations. Key improvements include:
>
> -We have unified the notation by defining r_i as the read-count values for both control and target counts, simplifying the expressions for L_PSR, L_LGR, L_con, and L_consist.
> -The definitions of M_i and R_i have been refined for clarity, and we explicitly define N as the training batch size in L_LGR and L_consist calculations.
> -Additionally, we now provide detailed expressions for \Delta D_ij and \Delta G_ij based on their constituent terms G_i and D_i.
>
> We have added explanations of how each component influences the ranking learning process and how their interactions contribute to robust read-count prediction. The revised section provides clear intuition for the multiplicative and subtractive terms in the objective function.
>
> > "a novel ranking loss that rectifies relative magnitude relationships between read counts enabling the learning of causal features determining activity levels". The use of 'causal' is particularly strong here. How do you disentangle correlation from causation? What aspects are explicitly causal?
>
> We appreciate the reviewer's critical observation regarding the term "causal." We acknowledge that our methodology identifies statistical associations between molecular features and activity levels, rather than establishing causation. We have revised the manuscript to use more precise language, now describing our approach as "a novel ranking loss that rectifies relative magnitude relationships between read counts, enabling the learning of features correlated with activity levels." This modification better reflects the statistical nature of our findings and avoids overstating the inferential capabilities of our method.

---

> ### Author Response · Authors · 2024-11-23
> **Response to Reviewer 55EM (continued)**
>
> > Was the pose encoder necessary? It seems superfluous to the other components added here, and quite complex.
>
> As addressed in our response to Question 2 (Experimental Setting), we acknowledge both the necessity and potential inaccuracy of pose information. To optimally utilize 3D structural information while mitigating potential noise, we implement a CNN-based pose encoder. This architecture, combined with an attention mechanism, enables the model to identify poses that strongly correlate with high binding affinity. The effectiveness of structural information is quantitatively demonstrated in our ablation study, with results shown below:
>
> **Parameter value comparison for structure scaling factor**
>
> *The best performance within one set of hyperparameter group is shown in bold.*
>
> | Value | 3p3h (CA2) | | 4kp5-A (CA12) | | 4kp5-OA (CA12) | |
> |-------|------------|------------|---------------|------------|----------------|------------|
> | Metric | Sp | SubSp | Sp | SubSp | Sp | SubSp |
> | 0 | -0.236±0.010 | -0.112±0.013 | -0.253±0.012 | -0.218±0.017 | -0.195±0.044 | -0.103±0.055 |
> | 0.3 | -0.262±0.008 | -0.145±0.012 | -0.265±0.017 | -0.227±0.017 | -0.124±0.146 | -0.062±0.090 |
> | 0.6 | -0.263±0.008 | -0.146±0.011 | -0.250±0.017 | -0.231±0.019 | -0.210±0.040 | -0.121±0.047 |
> | 1 | **-0.286±0.002** | **-0.177±0.005** | **-0.268±0.012** | **-0.277±0.016** | **-0.289±0.025** | **-0.233±0.021** |
> | 1.5 | -0.270±0.011 | -0.155±0.016 | -0.244±0.022 | -0.252±0.022 | -0.152±0.156 | -0.139±0.104 |
> | 2 | -0.271±0.012 | -0.155±0.015 | -0.191±0.089 | -0.216±0.060 | -0.230±0.038 | -0.152±0.051 |
>
>
> **Reference:**
>
> [1] Hou, R., Xie, C., Gui, Y., Li, G., & Li, X. (2023). Machine-learning-based data analysis method for cell-based selection of DNA-encoded libraries. ACS omega, 8(21), 19057-19071.
>
> [2] Koes, D. R., Baumgartner, M. P., & Camacho, C. J. (2013). Lessons learned in empirical scoring with smina from the CSAR 2011 benchmarking exercise. Journal of chemical information and modeling, 53(8), 1893-1904
>
> [3] Kuai, L., O’Keeffe, T., & Arico-Muendel, C. (2018). Randomness in DNA encoded library selection data can be modeled for more reliable enrichment calculation. SLAS DISCOVERY: Advancing the Science of Drug Discovery, 23(5), 405-416

---

> ### Author Response · Authors · 2024-11-25
> **Continuing Response to Reviewer 55EM**
>
> Dear Reviewer 55EM,
>
> Thank you very much for reviewing our paper! We really appreciate your time and effort in providing these helpful comments. We have done our best to address each point you raised and made improvements based on your suggestions.
>
> We hope you'll have a chance to take a look!

---

> ### Author Response · Authors · 2024-11-27
> **Continuing Response to Reviewer 55EM**
>
> Dear Reviewer 55EM,
>
> Following your valuable feedback, we have thoroughly revised the manuscript. If you find that we have adequately addressed your concerns, we would be grateful if you could consider adjusting your evaluation accordingly.
>
> Best regards,
>
> Authors  of Submission 1248

---

### Author Response · Authors · 2024-11-23
**Consolidated response to all reviewers and the AC**

# Response to Reviewers

We sincerely thank all reviewers and ACs for their invaluable feedback and constructive comments. Your insights have greatly helped us improve both the quality of our work and our perspective on the research problems. We deeply appreciate the time and effort you have invested in reviewing our manuscript.

We have carefully addressed all concerns and made substantial improvements to our paper. We hope our revisions adequately address your concerns while maintaining the core contributions of our work. We welcome you to reassess our paper in light of these changes and our detailed responses. The major revisions are:

## Article Presentation

### Introduction Section
- Restate current challenges in DEL denoising tasks
- Restate the main contributions of our methods
- State Activity labels and the construction process
- Introduce the connection between Activity Labels and Benzene Sulfonamide

### Methods Section
- Clarify the definition of ZIP (Zero-Inflated Poisson Distribution) and ZIP Loss
- Unify all related symbols including:
  - r_i, \hat r_i
  - y_i, \hat y_i
  - λ_M, λ_R
  - π_M, π_R
- Modify variable names in loss function definitions
- Edit depending variables in L_consist and L_con

### Experiments Section
- State Activity Label construction
- Clarify experimental settings of zero-shot experiment

## Experimental Setting

- Added hyperparameter weight ablation study on multiple datasets (3p3h, 4kp5-A, 4kp5-OA) in Appendix C.1
  - Validates DEL-Ranking's robustness to hyperparameters
- Added selection criteria for DEL-Ranking hyperparameters in Appendix B.2
- Added structure information ablation study on multiple datasets (3p3h, 4kp5-A, 4kp5-OA, 5fl4-9p, 5fl4-20p) in Appendix C.2
  - Validates effectiveness of structure information in identifying correlation between noisy read counts and compound Ki values
- Added detailed virtual docking details to dataset construction
- Added Top-50 candidate selection process of DEL-Dock for analyzing Top candidate detection ability

## Method Design

- Restated method motivation in Introduction Section
- Provided detailed motivation and design insight for all proposed techniques in Methods Section
- Added detailed description of ARC algorithm in Section 3.4
  - Provided insight on activity label effectiveness
  - Explained design logic of L_consist

In summary, we have thoroughly addressed all major concerns raised by the reviewers through substantial revisions and additional experiments. The expanded analyses on hyperparameter sensitivity, structure information ablation, and detailed methodological explanations have strengthened our work's technical depth and empirical validation. We have improved the clarity of our presentation through consistent notation, detailed motivation sections, and comprehensive experimental protocols. The additional results further validate our method's effectiveness and robustness across multiple datasets. We believe these revisions have significantly enhanced the quality and accessibility of our work while maintaining its novel contributions to DEL denoising research. We look forward to your feedback on our revised manuscript.

---

### Meta-Review · Area_Chair_C5GM · 2024-12-19

**Metareview:**

The paper tackles two main issues in modeling DEL-based data: distribution noise in low copy number regimes and the mismatch between read counts and true binding affinities. To do so, the authors introduced a new ranking loss and a self-training module to align read counts and binding affinities.  As an additional contribution. the author provide 3 curated DEL datasets for evaluation.

The paper addresses an interesting and relevant problem that is well-motivated and the approach promising. Unfortunately the original submission contained significant presentation issues and several essential details were missing (see e.g. concerns raised by Reviewer jJbZ and Reviewer 7ycA).

The AC thoroughly examined the original manuscript and its revised version, as well as all reviews and feedback. The authors have provided significant additional material and have made substantial modification to improve the manuscript, which improve completeness and readability. However, important concerns remain such as the use of inductive bias w.r.t benzene sulfonamide in the ARC loss as it relies on information that is not always available and that is not employed by baselines.

We recommend that the authors revise their manuscript to either assess their method in fully generality, i.e. (1) comparison with other approaches in cases where the information is not present (i.e. without ARC loss or with another way of constructing the ARC loss that does not rely on this information) and (2) a second set of results in cases where such information is available, with clear acknowledgement when it is not being used by some comparison approaches.

**Additional Comments On Reviewer Discussion:**

The reviewers were all concerned with the initial lack of clarity. The manuscript contained some errors and missing information that hindered full understanding of the significance of the contribution. In addition important details on side information being used by a key component of the proposed approach were missing, which restrict the generality of applicability of the approach and also makes the comparison with alternatives which do not employ such information less clear.

The authors provided significant feedback and have substantially revised their manuscript. Given the initial presentation issues (as raised by all reviewers), reviewer 7ycA suggested that the manuscript would benefit from a whole new round of reviews, which the AC agree with.

Also the AC believes that there is indeed a need to fully study the advantage of training on benzene sulfonamides to assess whether it is what yields to the performance gain as noted by Reviewer jJbZ.

---

### Decision · Program_Chairs · 2025-01-22

Reject